# Learning Augmented Energy Minimization via Speed Scaling

**Etienne Bamas**[*]
EPFL
Switzerland
etienne.bamas@epfl.ch

**Andreas Maggiori**[*]
EPFL
Switzerland
andreas.maggiori@epfl.ch

**Lars Rohwedder**[*]
EPFL
Switzerland
lars.rohwedder@epfl.ch

**Ola Svensson**[*]
EPFL
Switzerland
ola.svensson@epfl.ch

## Abstract

As power management has become a primary concern in modern data centers, computing resources are being scaled dynamically to minimize energy consumption. We initiate the study of a variant of the classic online *speed scaling* problem, in which machine learning predictions about the future can be integrated naturally. Inspired by recent work on learning-augmented online algorithms, we propose an algorithm which incorporates predictions in a black-box manner and outperforms any online algorithm if the accuracy is high, yet maintains provable guarantees if the prediction is very inaccurate. We provide both theoretical and experimental evidence to support our claims.

## 1 Introduction

Online problems can be informally defined as problems where we are required to make irrevocable decisions without knowing the future. The classical way of dealing with such problems is to design algorithms which provide provable bounds on the ratio between the value of the algorithm's solution and the optimal (offline) solution (the competitive ratio). Here, no assumption about the future is made. Unfortunately, this *no-assumption* regime comes at a high cost: Because the algorithm has to be overly prudent and prepare for all possible future events, the guarantees are often poor. Due to the success story of machine learning (ML), a recent line of work, first proposed by Lykouris and Vassilvitskii [13] and Medina and Vassilvitskii [14], suggests incorporating the predictions provided by ML algorithms in the design of online algorithms. While some related approaches were considered before (see e.g. Xu and Xu [16]), the attention in this subject has increased substantially in the recent years [7, 8, 10, 11, 12, 13, 14, 15]. An obvious caveat is that ML predictors often come with no worst-case guarantees and so we would like our algorithm to be robust to misleading predictions. We follow the terminology introduced by Purohit et al. [15], where consistency is the performance of an algorithm when the predictor is perfectly accurate, while robustness is a worst case guarantee that does not depend on the quality of the prediction. The goal of the works above is to design algorithms which provably beat the classical online algorithms in the consistency case, while being robust when the predictor fails.

**Problem.** The problem we are considering is motivated by the following scenario. Consider a server that receives requests in an online fashion. For each request some computational work has to

---

[*]Equal Contribution

be done and, as a measure of Quality-of-Service, we require that each request is answered within some fixed time. In order to satisfy all the requests in time the server can dynamically change its processor speed at any time. However, the power consumption can be a super-linear function of the processing speed (more precisely, we model the power consumption as $s^\alpha$ where $s$ is the processing speed and $\alpha > 1$). Therefore, the problem of minimizing energy becomes non-trivial. This problem can be considered in the online model where the server has no information about the future tasks at all. However, this assumption seems unnecessarily restrictive as these requests tend to follow some patterns that can be predicted. For this reason a good algorithm should be able to incorporate some given predictions about the future. Similar scenarios appear in real-world systems as, for instance, in dynamic frequency scaling of CPUs or in autoscaling of cloud applications [4, 9]. In the case of autoscaling, ML advice is already being incorporated into online algorithms in practice [4]. However, on the theory side, while the above speed scaling problem was introduced by Yao et al. [17] in a seminal paper who studied it both in the online and offline settings (see also [2, 3]), it has not been considered in the learning augmented setting.

**Contributions.** We formalize an intuitive and well-founded prediction model for the classic speed scaling problem. We show that our problem is non-trivial by providing an unconditional lower bound that demonstrates: An algorithm cannot be optimal, if the prediction is correct, and at the same time retain robustness. We then focus on our main contribution which is the design and analysis of a simple and efficient algorithm which incorporates any ML predictor as a black-box without making any further assumption. We achieve this in a modular way: First, we show that there is a consistent (but not robust) online algorithm. Then we develop a technique to make any online algorithm (which may use the prediction) robust at a small cost. Moreover, we design general methods to allow algorithms to cope with small perturbations in the prediction. In addition to the theoretical analysis, we also provide an experimental analysis that supports our claims on both synthetic and real datasets. For most of the paper we focus on a restricted case of the speed scaling problem by Yao et al. [17], where predictions can be integrated naturally. However, we show that with more sophisticated algorithms our techniques extend well to the general case.

**Related work.** On the one hand, the field of learning augmented algorithms is relatively new, with a lot of recent exciting results (see for example Gollapudi and Panigrahi [7], Hsu et al. [8], Kodialam [10], Lattanzi et al. [11], Lee et al. [12], Lykouris and Vassilvitskii [13], Medina and Vassilvitskii [14], Purohit et al. [15], Xu and Xu [16]). On the other hand, the speed scaling problem proposed by Yao et al. in [17] is well understood in both the offline and online setting. In its full generality, a set of tasks each with different arrival times, deadlines, and workloads needs to be completed in time while the speed is scaled in order to minimize energy. In the offline setting Yao et al. proved that the problem can be solved in polynomial time by a greedy algorithm. In the online setting, in which the jobs are revealed only at their release time, Yao et al. designed two different algorithms: (1) the AVERAGE RATE heuristic (AVR), for which they proved a bound of $2^{\alpha-1}\alpha^\alpha$ on the competitive ratio. This analysis was later proved to be asymptotically tight by Bansal et al. [3]. (2) The OPTIMAL AVAILABLE heuristic (OA), which was shown to be $\alpha^\alpha$-competitive in [2]. In the same paper, Bansal et al. proposed a third online algorithm named BKP for which they proved a competitive ratio asymptotically equivalent to $e^\alpha$. While these competitive ratios exponential in $\alpha$ might not seem satisfying, Bansal et al. also proved that the exponential dependency cannot be better than $e^\alpha$. A number of variants of the problem have also been considered in the offline setting (no preemption allowed, precedence constraints, nested jobs and more listed in a recent survey by Gerards et al. [6]) and under a stochastic optimization point of view (see for instance [1]). It is important to note that, while in theory the problem is interesting in the general case i.e. when $\alpha$ is an input parameter, in practice we usually focus on small values of $\alpha$ such as 2 or 3 since they model certain physical laws (see e.g. Bansal et al. [2]). Although the BKP algorithm provides the best asymptotic guarantee, OA or AVR often lead to better solutions for small $\alpha$ and therefore remain relevant.

## 2 Model and Preliminaries

We define the Uniform Speed Scaling problem, a natural restricted version of the speed scaling problem [17], where predictions can be integrated naturally. While the restricted version is our main focus as it allows for cleaner exposition and prediction models, we also show that our techniques

can be adapted to more complex algorithms yielding similar results for the general problem (see Section 3.4 for further extensions).

**Problem definition.** An instance of the problem can be formally described as a triple $(w, D, T)$ where $[0, T]$ is a finite time horizon, each time $i \in \{0, \dots, T - D\}$ jobs with a total workload $w_i \in \mathbb{Z}_{\geqslant 0}$ arrive, which have to be completed by time $i + D$. To do so, we can adjust the speed $s_i(t)$ at which each workload $w_i$ is processed for $t \in [i, i + D]$. Jobs may be processed in parallel. The overall speed of our processing unit at time $t$ is the sum $s(t) = \sum_i s_i(t)$, which yields a power consumption of $s(t)^\alpha$, where $\alpha > 1$ is a problem specific constant. Since we want to finish each job on time, we require that the amount of work dedicated to job $i$ in the interval $[i, i + D]$ should be $w_i$. In other words, $\int_i^{i+D} s_i(t) \, dt = w_i$. In the offline setting, the whole instance is known in advance, i.e., the vector of workloads $w$ is entirely accessible. In the online problem, at time $i$, the algorithm is only aware of all workloads $w_j$ with $j \leqslant i$, i.e., the jobs that were released before time $i$. As noted by Bansal et al. [2], in the offline setting the problem can be formulated concisely as the following mathematical program:

**Definition 1** (Uniform Speed Scaling problem). On input $(w, D, T)$ compute the optimal solution for

$$\min \int_0^T s(t)^\alpha \, dt \quad s.t. \ \forall i \ \int_i^{i+D} s_i(t) \, dt = w_i, \quad \forall t \ \sum_i s_i(t) = s(t), \quad \forall i \forall t \ s_i(t) \geqslant 0.$$

In contrast, we refer to the problem of Yao et al. [17] as the *General Speed Scaling* problem. The difference is that there the time that the processor is given to complete each job is not necessarily equal across jobs. More precisely, there we replace $w$ and $D$ by a set of jobs $J_j = (r_j, d_j, w_j)$, where $r_j$ is the time the job becomes available, $d_j$ is the deadline by which it must be completed, and $w_j$ is the work to be completed. As a shorthand, we sometimes refer to these two problems as the *uniform deadlines* case and the *general deadlines* case. As mentioned before, Yao et al. [17] provide a simple optimal greedy algorithm that runs in polynomial time. As for the online setting, we emphasize that both the general and the uniform speed scaling problem are non-trivial. More specifically, we prove that no online algorithm can have a competitive ratio better than $\Omega((6/5)^\alpha)$ even in the uniform case (see full version of the paper). We provide a few additional insights on the performance of online algorithms for the uniform deadline case. Although the AVR algorithm was proved to be $2^{\alpha-1} \cdot \alpha^\alpha$-competitive by Yao et al. [17] with a quite technical proof ; we show, with a simple proof, that AVR is in fact $2^\alpha$-competitive in the uniform deadlines case and we provide an almost matching lower bound on the competitive ratio (see full version of the paper).

Note that in both problems the processor is allowed to run multiple jobs in parallel. However, we underline that restricting the problem to the case where the processor is only allowed to run at most one job at any given point in time is equivalent. Indeed, given a feasible solution $s(t) = \sum_i s_i(t)$ in the parallel setting, rescheduling jobs sequentially according to the earliest deadline first (EDF) policy creates a feasible solution of the same (energy) cost where at each point in time only one job is processed.

**Prediction model and error measure.** In the following, we present the model of prediction we are considering. Recall an instance of the problem is defined as a time horizon $[0, T]$, a duration $D$, and a vector of workloads $w_i$, $i = 1, \dots, T - D$. A natural prediction is simply to give the algorithm a predicted instance $(w^{\text{pred}}, T, D)$ at time $t = 0$. From now on, we will refer to the ground truth work vector as $w^{\text{real}}$ and to the predicted instance as $w^{\text{pred}}$. We define the error err of the prediction as

$$\text{err}(w^{\text{real}}, w^{\text{pred}}) = ||w^{\text{real}} - w^{\text{pred}}||_\alpha^\alpha = \sum_i |w_i^{\text{real}} - w_i^{\text{pred}}|^\alpha.$$

We simply write err, when $w^{\text{real}}$ and $w^{\text{pred}}$ are clear from the context. The motivation for using $\alpha$ in the definition of err and not some other constant $p$ comes from strong impossibility results. Clearly, guarantees for higher values $p$ are weaker than for lower $p$. Therefore, we would like to set $p$ as low as possible. However, we show that $p$ needs to be at least $\alpha$ in order to make a sensible use of a prediction (see full version of the paper). We further note that it may seem natural to consider a predictor that is able to renew its prediction over time, e.g., by providing our algorithm a new prediction at every integral time $i$. To this end, in the full version of this paper, we show how to

naturally extend all our results from the single prediction to the evolving prediction model. Finally we restate some desirable properties previously defined in [13, 15] that a learning augmented algorithm should have. Recall that the prediction is a source of unreliable information on the remaining instance and that the algorithm is oblivious to the quality of this prediction. In the following we denote by OPT the energy cost of the optimal offline schedule and by $\varepsilon > 0$ a robustness parameter of the algorithm, the smaller $\varepsilon$ is the more we trust the prediction.

If the prediction is perfectly accurate, i.e., the entire instance can be derived from the prediction, then the provable guarantees should be better than what a pure online algorithm can achieve. Ideally, the algorithm produces an offline optimal solution or comes close to it. By close to optimal, we mean that the cost of the algorithm (when the prediction is perfectly accurate) should be at most $c(\alpha, \varepsilon) \cdot \text{OPT}$, where $c(\alpha, \varepsilon)$ tends to 1 as $\varepsilon$ approaches 0. This characteristic will be called **consistency**.

The competitive ratio of the algorithm should always be bounded even for arbitrarily bad (adversarial) predictions. Ideally, the competitive ratio is somewhat comparable to the competitive ratio of algorithms from literature for the pure online case. Formally, the cost of the algorithm should always be bounded by $r(\alpha, \varepsilon) \cdot \text{OPT}$ for some function $r(\alpha, \varepsilon)$. This characteristic will be called **robustness**.

A perfect prediction is a strong requirement. The consistency property should transition smoothly for all ranges of errors, that is, the algorithm's guarantees deteriorate smoothly as the prediction error increases. Formally, the cost of the algorithm should always be at most $c(\alpha, \varepsilon) \cdot \text{OPT} + f(\alpha, \varepsilon, \text{err})$ for some function $f$ such that $f(\alpha, \varepsilon, 0) = 0$ for any $\alpha, \varepsilon$. This last property will be called **smoothness**.

Note that our definitions of consistency and robustness depend on the problem specific constant $\alpha$ which is unavoidable (see full version of the paper). The dependence on the robustness parameter $\varepsilon$ is justified, because no algorithm can be perfectly consistent and robust at the same time (see full version), hence a trade-off is necessary.

## 3 Algorithm

In this section we develop two modular building blocks to obtain a consistent, smooth, and robust algorithm. The first block is an algorithm which computes a schedule online taking into account the prediction for the future. This algorithm is consistent and smooth, but not robust. Then we describe a generic method how to robustify an arbitrary online algorithm at a small cost. Finally, we give a summary of the theoretical qualities for the full algorithm and a full description in pseudo-code. We note that in the full version of the paper we present additional building blocks (see Section 3.4 for an overview).

### 3.1 A Consistent and Smooth Algorithm

In the following we describe a learning-augmented online algorithm, which we call LAS-TRUST.

**Preparation.** We compute an optimal schedule $s^{\text{pred}}$ for the predicted jobs. An optimal schedule can always be normalized such that each workload $w_i^{\text{pred}}$ is completely scheduled in an interval $[a_i, b_i]$ at a uniform speed $c_i$, that is,

$$s_i^{\text{pred}}(t) = \begin{cases} c_i & \text{if } t \in [a_i, b_i], \\ 0 & \text{otherwise.} \end{cases}$$

Furthermore, the intervals $[a_i, b_i]$ are non-overlapping. For details we refer the reader to the optimal offline algorithm by Yao et al. [17], which always creates such a schedule.

**The online algorithm.** At time $i$ we first schedule $w_i^{\text{real}}$ at uniform speed in $[a_i, b_i]$, but we cap the speed at $c_i$. If this does not complete the job, that is, $w_i^{\text{real}} > c_i(b_i - a_i) = w_i^{\text{pred}}$, we uniformly schedule the remaining work in the interval $[i, i + D]$

More formally, we define $s_i(t) = s_i'(t) + s_i''(t)$ where

$$s_i'(t) = \begin{cases} \min\left\{\frac{w_i^{\text{real}}}{b_i - a_i}, c_i\right\} & \text{if } t \in [a_i, b_i], \\ 0 & \text{otherwise.} \end{cases}$$

and

$$s_i''(t) = \begin{cases} \frac{1}{D} \max\{0, w_i^{\text{real}} - w_i^{\text{pred}}\} & \text{if } t \in [i, i + D], \\ 0 & \text{otherwise.} \end{cases}$$

**Analysis.** It is easy to see that the algorithm is consistent: If the prediction of $w_i^{\text{real}}$ is perfect ($w_i^{\text{pred}} = w_i^{\text{real}}$), the job will be scheduled at speed $c_i$ in the interval $[a_i, b_i]$. If all predictions are perfect, this is exactly the optimal schedule.

**Theorem 2.** *For every $0 < \delta \leqslant 1$, the cost of the schedule produced by the algorithm* LAS-TRUST *is bounded by $(1 + \delta)^\alpha \text{OPT} + (12/\delta)^\alpha \cdot \text{err}$.*

*Proof.* Define $w_i^+ = \max\{0, w_i^{\text{real}} - w_i^{\text{pred}}\}$ as the additional work at time $i$ as compared to the predicted work. Likewise, define $w_i^- = \max\{0, w_i^{\text{pred}} - w_i^{\text{real}}\}$. We use $\text{OPT}(w^+)$ and $\text{OPT}(w^-)$ to denote the cost of optimal schedules of these workloads $w^+$ and $w^-$, respectively. We will first relate the energy of the schedule $s(t)$ to the optimal energy for the predicted instance, i.e., $\text{OPT}(w^{\text{pred}})$. Then we will relate $\text{OPT}(w^{\text{pred}})$ to $\text{OPT}(w^{\text{real}})$.

For the former let $s_i'$ and $s_i''$ be defined as in the algorithm. Observe that $s_i'(t) \leqslant s_i^{\text{pred}}(t)$ for all $i$ and $t$. Hence, the energy for the partial schedule $s'$ (by itself) is at most $\text{OPT}(w^{\text{pred}})$. Furthermore, by definition we have that $s_i''(t) = w_i^+/D$. In other words, $s''$ is exactly the AVR schedule on instance $w^+$. By analysis of AVR, we know that the total energy of $s_i''$ is at most $2^\alpha \text{OPT}(w^+)$. Since the energy function is non-linear, we cannot simply add the energy of both speeds. Instead, we use the following inequality: For all $x, y \geqslant 0$ and $0 < \gamma \leqslant 1$, it holds that $(x + y)^\alpha \leqslant (1 + \gamma)^\alpha x^\alpha + \left(\frac{2}{\gamma}\right)^\alpha y^\alpha$. This follows from a simple case distinction whether $y \leqslant \gamma x$. Thus, (substituting $\gamma$ for $\delta/3$) the energy of the schedule $s$ is bounded by

$$\int (s'(t) + s''(t))^\alpha dt \leqslant (1 + \delta/3)^\alpha \int s_i'(t)^\alpha dt + (6/\delta)^\alpha \int s_i''(t)^\alpha dt$$

$$\leqslant (1 + \delta/3)^\alpha \text{OPT}(w^{\text{pred}}) + (12/\delta)^\alpha \text{OPT}(w^+). \quad (1)$$

For the last inequality we used that the competitive ratio of AVR is $2^\alpha$.

In order to relate $\text{OPT}(w^{\text{pred}})$ and $\text{OPT}(w^{\text{real}})$, we argue similarly. Notice that scheduling $w^{\text{real}}$ optimally (by itself) and then scheduling $w^-$ using AVR forms a valid solution for $w^{\text{pred}}$. Hence,

$$\text{OPT}(w^{\text{pred}}) \leqslant (1 + \delta/3)^\alpha \text{OPT}(w^{\text{real}}) + (12/\delta)^\alpha \text{OPT}(w^-).$$

Inserting this inequality into (1) we conclude that the energy of the schedule $s$ is at most

$$(1 + \delta/3)^{2\alpha} \text{OPT}(w^{\text{real}}) + (12/\delta)^\alpha (\text{OPT}(w^+) + \text{OPT}(w^-))$$

$$\leqslant (1 + \delta)^\alpha \text{OPT}(w^{\text{real}}) + (12/\delta)^\alpha \cdot \text{err}.$$

This inequality follows from the fact that the error function $\|\cdot\|_\alpha^\alpha$ is always an upper bound on the energy of the optimal schedule (by scheduling every job within the next time unit). □

### 3.2 Robustification

In this section, we describe a method ROBUSTIFY that takes any online algorithm which guarantees to complete each job in $(1 - \delta)D$ time, that is, with some slack to its deadline, and turns it into a robust algorithm without increasing the energy of the schedule produced. Here $\delta > 0$ can be chosen at will, but it impacts the robustness guarantee. We remark that the slack constraint is easy to achieve: In the full version of the paper we prove that decreasing $D$ to $(1 - \delta)D$ increases the energy of the optimum schedule only very mildly. Specifically, if we let $\text{OPT}(w^{\text{real}}, (1 - \delta)D, T)$ and $\text{OPT}(w^{\text{real}}, D, T)$ denote the costs of optimal schedules of workload $w^{\text{real}}$ with durations $(1 - \delta)D$ and $D$, respectively, then:

**Claim 3.** *For any instance $(w^{\text{real}}, D, T)$ we have that $\text{OPT}(w^{\text{real}}, (1 - \delta)D, T) \leqslant \frac{1}{(1-\delta)^{\alpha-1}} \text{OPT}(w^{\text{real}}, D, T)$.*

Hence, running a consistent algorithm with $(1 - \delta)D$ will not increase the cost significantly. Alternatively, we can run the online algorithm with $D$, but increase the generated speed function by $1/(1 - \delta)$

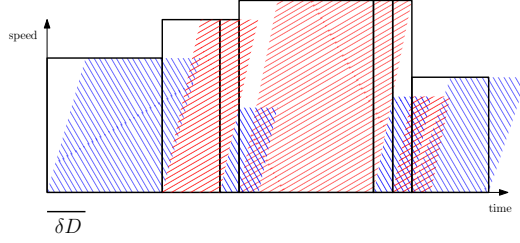

Figure 1: A schedule and its convolution.

and reschedule all jobs using EDF. This also results in a schedule where all jobs are completed in $(1 - \delta)D$ time.

For a schedule $s$ of $(w^{\text{real}}, (1 - \delta)D, T)$ we define the $\delta$-convolution operator which returns the schedule $s^{(\delta)}$ of the original instance $(w^{\text{real}}, D, T)$ by

$$s_i^{(\delta)}(t) = \frac{1}{\delta D} \int_{t-\delta D}^{t} s_i(r) \, dr$$

for each $i \in T$ (letting $s_i(r) = 0$ if $r < 0$). See Figure 1 for an illustration. The name comes from the fact that this operator is the convolution of $s_i(t)$ with the function $f(t)$ that takes value $1/(\delta D)$ if $0 \leqslant t \leqslant \delta D$ and value $0$ otherwise.

Next we state three key properties of the convolution operator, all of which follow from easy observations or standard arguments that are deferred to the full version of the paper.

**Claim 4.** *If $s$ is a feasible schedule for $(w^{\text{real}}, (1 - \delta)D, T)$ then $s^{(\delta)}$ is a feasible schedule for $(w^{\text{real}}, D, T)$.*

**Claim 5.** *The cost of schedule $s^{(\delta)}$ is not higher than that of $s$, that is,*

$$\int_0^T (s^{(\delta)}(t))^\alpha dt \leqslant \int_0^T (s(t))^\alpha dt.$$

Let $s_i^{\text{AVR}}(t)$ denote the speed of workload $w_i^{\text{real}}$ of the AVERAGE RATE heuristic, that is, $s_i^{\text{AVR}}(t) = w_i^{\text{real}}/D$ if $i \leqslant t \leqslant i + D$ and $s_i^{\text{AVR}}(t) = 0$ otherwise. We relate $s_i^{(\delta)}(t)$ to $s_i^{\text{AVR}}(t)$.

**Claim 6.** *Let $s$ be a feasible schedule for $(w^{\text{real}}, (1 - \delta)D, T)$. Then $s_i^{(\delta)}(t) \leqslant \frac{1}{\delta} s_i^{\text{AVR}}(t)$.*

By using that the competitive ratio of AVERAGE RATE is at most $2^\alpha$ (see the full version of this paper), we get

$$\int_0^T (s^{(\delta)}(t))^\alpha dt \leqslant \left(\frac{1}{\delta}\right)^\alpha \int_0^T (s^{\text{AVR}}(t))^\alpha dt \leqslant \left(\frac{2}{\delta}\right)^\alpha \text{OPT} .$$

We conclude with the following theorem, which follows immediately from the previous claims.

**Theorem 7.** *Given an online algorithm that produces a schedule $s$ for $(w^{\text{real}}, (1 - \delta)D, T)$, we can compute online a schedule $s^{(\delta)}$ with*

$$\int_0^T (s^{(\delta)}(t))^\alpha dt \leqslant \min \left\{ \int_0^T (s(t))^\alpha dt, \ \left(\frac{2}{\delta}\right)^\alpha \text{OPT} \right\}.$$

### 3.3 Summary of the Algorithm

By combining LAS-TRUST and ROBUSTIFY, we obtain an algorithm LAS (see Algorithm 1) which has the following properties. See the full version of this paper for a formal argument.

**Theorem 8.** *For any given $\varepsilon > 0$, algorithm LAS constructs a schedule of cost at most $\min \left\{ (1 + \varepsilon) \text{OPT} + O\left(\frac{\alpha}{\varepsilon}\right)^\alpha \text{err}, \ O\left(\frac{\alpha}{\varepsilon}\right)^\alpha \text{OPT} \right\}.$*

---

**Algorithm 1** LEARNING AUGMENTED SCHEDULING (LAS)

---

**Input:** $T$, $D$, and $w^{\text{pred}}$ initially and $w^{\text{real}}$ in an online fashion
**Output:** A feasible schedule $(s_i)_{i=0}^{T-D}$
Let $\delta > 0$ with $\left(\frac{1+\delta}{1-\delta}\right)^{\alpha} = 1 + \varepsilon$.

Compute optimal offline schedule for $(w^{\text{pred}}, T, (1-\delta)D)$ where the jobs $w_i^{\text{pred}}$ are run at uniform speeds $c_i$ an disjoint intervals $[a_i, b_i]$ using [17].

**on arrival of** $w_i^{\text{real}}$ **do**

$\quad$ Let $s_i'(t) = \begin{cases} \min\left\{\frac{w_i^{\text{real}}}{b_i - a_i}, c_i\right\} & \text{if } t \in [a_i, b_i], \\ 0 & \text{otherwise.} \end{cases}$

$\quad$ Let $s_i''(t) = \begin{cases} \frac{1}{D}\max\{0, w_i^{\text{real}} - w_i^{\text{pred}}\} & \text{if } t \in [i, i + D], \\ 0 & \text{otherwise.} \end{cases}$

$\quad$ Let $s_i(t) = \frac{1}{\delta D}\int_{t-\delta D}^{t} s_i'(r) + s_i''(r)\, dr$
**end on**

---

## 3.4 Other Extensions

In the full version we also consider General Speed Scheduling (the problem with general deadlines) and show that a more sophisticated method allows us to robustify any algorithm even in this more general setting. Hence, for this case we can also obtain an algorithm that is almost optimal in the consistency case and always robust.

The careful reader may have noted that one can craft instances so that the used error function $\text{err}$ is very sensitive to small shifts in the prediction. An illustrative example is as follows. Consider a predicted workload $w^{\text{pred}}$ defined by $w_i^{\text{pred}} = 1$ for those time steps $i$ that are divisible by a large constant, say 1000, and let $w_i^{\text{pred}} = 0$ for all other time steps. If the real instance $w^{\text{real}}$ is a small shift of $w^{\text{pred}}$ say $w_{i+1}^{\text{real}} = w_i^{\text{pred}}$ then the prediction error $\text{err}(w^{\text{real}}, w^{\text{pred}})$ is large although $w^{\text{pred}}$ intuitively forms a good prediction of $w^{\text{real}}$. To overcome this sensitivity, we first generalize the definition of $\text{err}$ to $\text{err}_\eta$ which is tolerant to small shifts in the workload. In particular, $\text{err}_\eta(w^{\text{real}}, w^{\text{pred}}) = 0$ for the example given above. We then give a generic method for transforming an algorithm so as to obtain guarantees with respect to $\text{err}_\eta$ instead of $\text{err}$ at a small loss. Details can be found in the full version of the paper.

## 4 Experimental analysis

In this section, we will test the LAS algorithm on both synthetic and real datasets. We will calculate the competitive ratios with respect to the offline optimum. We fix $\alpha = 3$ in all our experiments as this value models the power consumption of modern processors (see Bansal et al. [2]). For each experiment, we compare our LAS algorithm to the three main online algorithms that exist for this problem which are AVR and OA by Yao et al. [17] and BKP by Bansal et al. [2]. We note that the code is publicly available at `https://github.com/andreasr27/LAS`.

**Artificial datasets.** In the synthetic data case, we will mimic the request pattern of a typical data center application by simulating a bounded random walk. In the following we write $Z \sim \mathcal{U}\{m, M\}$ when sampling an integer uniformly at random in the range $[m, M]$. Subsequently, we fix three integers $s, m, M$ where $[m, M]$ define the range in which the walk should stay. For each integral time $i$ we sample $X_i \sim \mathcal{U}\{-s, s\}$. Then we set $w_0 \sim \mathcal{U}\{m, M\}$ and $w_{i+1}$ to be the median value of the list $\{m, w_i + X_i, M\}$, that is, if the value $w_i + X_i$ remains in the predefined range we do not change it, otherwise we round it to the closest point in the range. For this type of ground truth instance we test our algorithm coupled with three different predictors. The **accurate** predictor for which we set $\tilde{w}_i \sim w_i + \mathcal{U}\{-s, s\}$, the **random** predictor where we set $\tilde{w}_i \sim \mathcal{U}\{m, M\}$ and the **misleading** predictor for which $\tilde{w}_i = (M - w_i) + m$. In each case we perform 20 experiment runs. The results are summarized in Table 1. In the first two cases (accurate and random predictors) we present the average competitive ratios of every algorithm over all runs. In contrast, for the last column

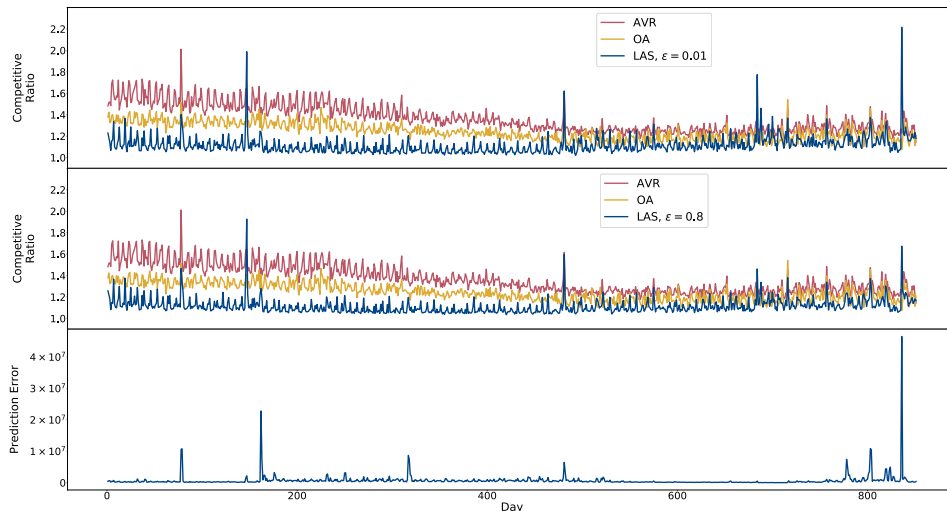

Figure 2: From top to bottom: The first two graphs show the performance of LAS when $\varepsilon = 0.01$ and $\varepsilon = 0.8$ with respect to the online algorithms AVR and OA. The bottom graph presents the prediction error. The timeline was discretized in chunks of ten minutes and $D$ was set to 20.

Table 1: Artificial dataset results

| Algorithm | Accurate | Random | Misleading |
|---|---|---|---|
| AVR | 1.268 | 1.268 | 1.383 |
| BKP | 7.880 | 7.880 | 10.380 |
| OA | 1.199 | 1.199 | 1.361 |
| LAS, $\varepsilon = 0.8$ | 1.026 | 1.203 | 1.750 |
| LAS, $\varepsilon = 0.6$ | 1.022 | 1.207 | 1.758 |
| LAS, $\varepsilon = 0.4$ | 1.018 | 1.213 | 1.767 |
| LAS, $\varepsilon = 0.2$ | 1.013 | 1.224 | 1.769 |
| LAS, $\varepsilon = 0.01$ | 1.008 | 1.239 | 1.766 |

We used $m = 20$, $M = 80$, $s = 5$, $T = 220$ and $D = 20$.

(misleading predictor) we present the maximum competitive ratio of each algorithm taken over the 20 runs to highlight the worst case robustness of LAS. We note that in the first case, where the predictor is relatively accurate but still noisy, LAS is consistently better than any online algorithm achieving a competitive ratio close to 1 for small values of $\varepsilon$. In the second case, the predictor does not give us useful information about the future since it is completely uncorrelated with the ground truth instance. In such a case, LAS experiences a similar performance to the best online algorithms. In the third case, the predictor tries to mislead our algorithm by creating a prediction which constitutes a symmetric (around $(m + M)/2$) random walk with respect to the true instance. When coupled with such a predictor, as expected, LAS performs worse than the best online algorithm, but it still maintains an acceptable competitive ratio. Furthermore, augmenting the robustness parameter $\varepsilon$, and thereby trusting less the predictor, improves the competitive ratio in this case.

**Real dataset.** We provide additional evidence that the LAS algorithm outperforms purely online algorithms by conducting experiments on the login requests to *BrightKite* [5], a no longer functioning social network. We note that this dataset was previously used in the context of learning augmented algorithms by Lykouris and Vassilvitskii [13]. In order to emphasize the fact that even a very simple predictor can improve the scheduling performance drastically, we will use the arguably most simple predictor possible. We use the access patterns of the previous day as a prediction for the current day. In Figure 2 we compare the performance of the LAS algorithm for different values of the robustness parameter $\varepsilon$ with respect to AVR and OA. We did not include BKP, since its performance is substantially worse than all other algorithms. Note that our algorithm shows a substantial improvement with respect to both AVR and OA, while maintaining a low competitive

ratio even when the prediction error is high (for instance in the last days). The first 100 days, where the prediction error is low, by setting $\varepsilon = 0.01$ (and trusting more the prediction) we obtain an average competitive ratio of 1.134, while with $\varepsilon = 0.8$ the average competitive ratio slightly deteriorates to 1.146. However, when the prediction error is high, setting $\varepsilon = 0.8$ is better. On average from the first to the last day of the timeline, the competitive ratio of AVR and OA is 1.36 and 1.24 respectively, while LAS obtains an average competitive ratio of 1.116 when $\varepsilon = 0.01$ and 1.113 when $\varepsilon = 0.8$, thus beating the online algorithms in both cases.

More experiments regarding the influence of the $\alpha$ parameter in the performance of LAS algorithm can be found in the full version of the paper.

## Broader impact

As climate change is a severe issue, trying to minimize the environmental impact of modern computer systems has become a priority. High energy consumption and the $CO_2$ emissions related to it are some of the main factors increasing the environmental impact of computer systems. While our work considers a specific problem related to scheduling, we would like to emphasize that a considerable percentage of real-world systems already have the ability to dynamically scale their computing resources[2] to minimize their energy consumption. Thus, studying models (like the one presented in this paper) with the latter capability is a line of work with huge potential societal impact. In addition to that, although the analysis of the guarantees provided by our algorithm is not straightforward, the algorithm itself is relatively simple. The latter fact makes us optimistic that insights from this work can be used in practice contributing to minimizing the environmental impact of computer infrastructures.

## Acknowledgments and Disclosure of Funding

This research is supported by the Swiss National Science Foundation project 200021-184656 "Randomness in Problem Instances and Randomized Algorithms". Andreas Maggiori was supported by the Swiss National Science Fund (SNSF) grant nᵒ 200020_182517/1 "Spatial Coupling of Graphical Models in Communications, Signal Processing, Computer Science and Statistical Physics".

## Footnotes

[2]CPU Dynamic Voltage and Frequency Scaling (DVFS) in modern processors and autoscaling of cloud applications

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
