[Supplementary Material]

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

For the former let $s_i'$ and $s_i''$ be defined as in the algorithm. Observe that $s_i'(t) \leqslant s_i^{\mathrm{pred}}(t)$ for all $i$ and $t$. Hence, the energy for the partial schedule $s'$ (by itself) is at most $\mathrm{OPT}(w^{\mathrm{pred}})$. Furthermore, by definition we have that $s_i''(t) = w_i^+/D$. In other words, $s_i''$ is exactly the AVR schedule on instance $w^+$. By analysis of AVR, we know that the total energy of $s_i''$ is at most $2^\alpha \, \mathrm{OPT}(w^+)$. Since the energy function is non-linear, we cannot simply add the energy of both speeds. Instead, we use the following inequality: For all $x, y \geqslant 0$ and $0 < \gamma \leqslant 1$, it holds that $(x + y)^\alpha \leqslant (1 + \gamma)^\alpha x^\alpha + \left(\frac{2}{\gamma}\right)^\alpha y^\alpha$. This follows from a simple case distinction whether $y \leqslant \gamma x$. Thus, (substituting $\gamma$ for $\delta/3$) the energy of the schedule $s$ is bounded by

$$\int (s'(t) + s''(t))^\alpha dt \leqslant (1 + \delta/3)^\alpha \int s_i'(t)^\alpha dt + (6/\delta)^\alpha \int s_i''(t)^\alpha dt$$
$$\leqslant (1 + \delta/3)^\alpha \, \mathrm{OPT}(w^{\mathrm{pred}}) + (12/\delta)^\alpha \, \mathrm{OPT}(w^+). \tag{1}$$

For the last inequality we used that the competitive ratio of AVR is $2^\alpha$.

In order to relate $\mathrm{OPT}(w^{\mathrm{pred}})$ and $\mathrm{OPT}(w^{\mathrm{real}})$, we argue similarly. Notice that scheduling $w^{\mathrm{real}}$ optimally (by itself) and then scheduling $w^-$ using AVR forms a valid solution for $w^{\mathrm{pred}}$. Hence,

$$\mathrm{OPT}(w^{\mathrm{pred}}) \leqslant (1 + \delta/3)^\alpha \, \mathrm{OPT}(w^{\mathrm{real}}) + (12/\delta)^\alpha \, \mathrm{OPT}(w^-).$$

Inserting this inequality into (1) we conclude that the energy of the schedule $s$ is at most

$$(1 + \delta/3)^{2\alpha} \, \mathrm{OPT}(w^{\mathrm{real}}) + (12/\delta)^\alpha (\mathrm{OPT}(w^+) + \mathrm{OPT}(w^-))$$
$$\leqslant (1 + \delta)^\alpha \, \mathrm{OPT}(w^{\mathrm{real}}) + (12/\delta)^\alpha \cdot \mathrm{

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

# A  Omitted Proofs from Section 3

**Theorem 8.** *For any given $\varepsilon > 0$, algorithm* LAS *constructs a schedule of cost at most* $\min \left\{ (1 + \varepsilon)\,\mathrm{OPT} + O\left(\frac{\alpha}{\varepsilon}\right)^{\alpha} \mathrm{err},\ O\left(\frac{\alpha}{\varepsilon}\right)^{\alpha} \mathrm{OPT} \right\}.$

*Proof.* We choose $\delta$ such that $(\frac{1+\delta}{1-\delta})^{\alpha} = 1 + \varepsilon$. Note that $\delta \leqslant \varepsilon/(6\alpha)$. By Claim 3 we know that

$$\mathrm{OPT}(w^{\mathrm{real}}, (1-\delta)D, T) \leqslant \left(\frac{1}{1-\delta}\right)^{\alpha} \mathrm{OPT}.$$

Hence, by Theorem 2 algorithm LAS-TRUST constructs a schedule with cost at most

$$\left(\frac{1+\delta}{1-\delta}\right)^{\alpha} \mathrm{OPT} + O\left(\frac{1}{\delta}\right)^{\alpha} \mathrm{err}$$

Finally, we apply ROBUSTIFY and with Theorem 7 obtain a bound of

$$\min \left\{ \left(\frac{1+\delta}{1-\delta}\right)^{\alpha} \mathrm{OPT} + O\left(\frac{1}{\delta}\right)^{\alpha} \mathrm{err},\ O\left(\frac{1}{\delta}\right)^{\alpha} \mathrm{OPT} \right\}$$

$$\leqslant \min \left\{ (1 + \varepsilon)\,\mathrm{OPT} + O\left(\frac{\alpha}{\varepsilon}\right)^{\alpha} \mathrm{err},\ O\left(\frac{\alpha}{\varepsilon}\right)^{\alpha} \mathrm{OPT} \right\}. \quad \square$$

# B  Pure online algorithms for uniform deadlines

Since most related results concern the general speed scaling problem, we give some insights about the uniform speed scaling problem in the online setting without predictions. We first give a lower bound on the competitive ratio for any online algorithm for the simplest case where $D = 2$ and then provide an almost tight analysis of the competitive ratio of AVR.

**Theorem 9.** *There is no (randomized) online algorithm with an (expected) competitive ratio better than* $\Omega\left((6/5)^{\alpha}\right).$

*Proof.* Consider $D = 2$ and two instances $J_1$ and $J_2$. Instance $J_1$ consists of only one job that is released at time 0 with workload 1 and $J_2$ consists of the same first job with a second job which starts at time 1 with workload 2.

In both instances, the optimal schedule runs with uniform speed at all time. In the first instance, it runs the single job for 2 units of time at speed $1/2$. The energy-cost is therefore $1/2^{\alpha-1}$. In the second instance, it first runs the first job at speed 1 for one unit of time and then the second job at speed 1 for 2 units of time. Hence, it has an energy-cost of 3.

Now consider an online algorithm. Before time 1 both instances are identical and the algorithm therefore behaves the same. In particular, it has to decide how much work of job 1 to process between time 0 and 1. Let us fix some $\gamma \geqslant 0$ as a threshold for the amount of work dedicated to job 1 by the algorithm before time 1. We have the following two cases depending on the instance.

1. If the algorithm processes more that $\gamma$ units of work on job 1 before time 1 then for instance $J_1$ the energy cost is at least $\gamma^{\alpha}$. Hence the competitive ratio is at least $\gamma^{\alpha} \cdot 2^{\alpha-1}$.

2. On the contrary, if the algorithm works less than $\gamma$ units of work before the release of the second job then in instance $J_2$ the algorithm has to complete at least $3 - \gamma$ units of work between time 1 and 3. Hence, its competitive ratio is at least $2/3 \cdot ((3 - \gamma)/2)^{\alpha}$.

Choosing $\gamma$ such that these two competitive ratios are equal gives $\gamma = \frac{3}{3^{1/\alpha}4^{1-1/\alpha}+1}$ and yields a lower bound on the competitive ratio of at least:

$$2^{\alpha-1} \left(\frac{3}{3^{1/\alpha}4^{1-1/\alpha} + 1}\right)^{\alpha}.$$

This term asymptotically approaches $1/2 \cdot (6/5)^{\alpha}$ and this already proves the theorem for deterministic algorithms. More precisely, it proves that any deterministic algorithm has a competitive ratio of

at least $\Omega\left((6/5)^\alpha\right)$ on at least one of the two instances $J_1$ or $J_2$. Hence, by defining a probability distribution over inputs such that $p(J_1) = p(J_2) = \frac{1}{2}$ and applying Yao's minimax principle we get that the expected competitive ratio of any randomized online algorithm is at least

$$(1/2) \cdot 2^{\alpha-1} \left(\frac{3}{3^{1/\alpha}4^{1-1/\alpha} + 1}\right)^\alpha .$$

which again gives $\Omega\left((6/5)^\alpha\right)$ as lower bound, this time against randomized algorithms. $\qquad \square$

We now turn ourselves to the more specific case of the AVR algorithm with the following two results. We recall that the AVR algorithm was shown to be $2^{\alpha-1} \cdot \alpha^\alpha$-competitive by Yao et al. [17] in the general deadlines case. In the case of uniform deadlines, the competitive ratio of AVR is actually much better and proofs are much less technical than the original analysis of Yao et al. Recall that for each job $i$ with workload $w_i$, release $r_i$, and deadline $d_i$ ; AVR defines a speed $s_i(t) = w_i/(d_i - r_i)$ if $t \in [r_i, d_i]$ and 0 otherwise.

**Theorem 10.** *AVR is $2^\alpha$-competitive for the uniform speed scaling problem.*

*Proof.* Let $(w, D, T)$ be a job instance and $s_{\text{OPT}}$ be the speed function of the optimal schedule for this instance. Let $s_{\text{AVR}}$ be the speed function produced by the AVERAGE RATE heuristic on the same instance. It suffices to show that for any time $t$ we have

$$s_{\text{AVR}}(t) \leqslant 2 \cdot s_{\text{OPT}}(t).$$

Fix some $t$. We assume w.l.o.g. that the optimal schedule runs each job $j$ isolated for a total time of $p_j^*$. By optimality of the schedule, the speed during this time is uniform, i.e., exactly $w_j/p_j^*$. Denote by $j_t$ the job that is processed in the optimal schedule at time $t$.

Let $j$ be some job with $r_j \leqslant t \leqslant r_j + D$. It must be that

$$\frac{w_j}{p_j^*} \leqslant \frac{w_{j_t}}{p_{j_t}^*} = s_{\text{OPT}}(t). \tag{2}$$

Note that all jobs $j$ with $r_j \leqslant t \leqslant r_j + D$ are processed completely between $t - D$ and $t + D$. Therefore,

$$\sum_{J_j : r_j \leqslant t \leqslant r_j + D} p_j^* \leqslant 2D.$$

With (2) it follows that

$$\sum_{J_j : r_j \leqslant t \leqslant r_j + D} w_j \leqslant s_{\text{OPT}}(t) \sum_{J_j : r_j \leqslant t \leqslant r_j + D} p_j^* \leqslant 2D \cdot s_{\text{OPT}}(t).$$

We conclude that

$$s_{\text{AVR}}(t) = \sum_{J_j : r_j \leqslant t \leqslant r_j + D} \frac{w_j}{D} \leqslant 2 \cdot s_{\text{OPT}}(t). \qquad \square$$

Next, we show that our upper bound on the exponential dependency in $\alpha$ of the competitive ratio for AVR (in Theorem 10) is tight for the uniform deadlines case.

**Theorem 11.** *Asymptotically ($\alpha$ approaches $\infty$), the competitive ratio of the AVR algorithm for the uniform deadlines case is at least*

$$\frac{2^\alpha}{e\alpha}$$

*Proof.* Assume $\alpha > 2$ and consider a two-job instance with one job arriving at time 0 of workload 1 and one job arriving at time $(1 - 2/\alpha)D$ with workload 1. One can check that the optimal schedule runs at constant speed throughout the whole instance for a total energy of

$$\left(\frac{2}{(2 - 2/\alpha)D}\right)^\alpha \cdot (2 - 2/\alpha)D.$$

On the other hand, on interval $[(1 - 2/\alpha)D, D]$, AVR runs at speed $2/D$. This implies the following lower bound on the competitive ratio:

$$\frac{(2/D)^\alpha \cdot (2/\alpha)D}{\left(\frac{2}{(2-2/\alpha)D}\right)^\alpha \cdot (2 - 2/\alpha)D} = \frac{2^\alpha}{\alpha}\left(1 - \frac{1}{\alpha}\right)^{\alpha-1}$$

which approaches to $2^\alpha/(e\alpha)$ as $\alpha$ tends to infinity. $\qquad\square$

## C  Impossibility results for learning augmented speed scaling

This section is devoted to prove some impossibility results about learning augmented algorithms in the context of speed scaling. We first prove that our trade-offs between consistency and robustness are essentially optimal. Again, we describe an instance as a triple $(w, D, T)$.

**Theorem 12.** *Assume a deterministic learning enhanced algorithm is $(1 + \varepsilon/3)^{\alpha-1}$-consistent for any $\alpha \geqslant 1$ and any small enough constant $\varepsilon > 0$ (independently of $D$). Then the worst case competitive ratio of this algorithm cannot be better than $\Omega\left(\frac{1}{\varepsilon}\right)^{\alpha-1}$.*

*Proof.* Fix $D$ big enough so that $\lceil \varepsilon D \rceil \leqslant 2 \cdot (\varepsilon D)$. Consider two different job instances $J_1$ and $J_2$: $J_1$ contains only one job of workload 1 released at time 0 and $J_2$ contains an additional job of workload $1/\varepsilon$ released at time $\lceil \varepsilon D \rceil$. On the first instance, the optimal cost is $1/D^{\alpha-1}$ while the optimum energy cost for $J_2$ is $(1/\lceil \varepsilon D \rceil)^{\alpha-1} + D/(\varepsilon D)^\alpha \leqslant (1/\varepsilon)^\alpha \cdot ((1 + \varepsilon)/D^{\alpha-1})$.

Assume the algorithm is given the job of workload 1 released at time 0 and additionally the prediction consists of one job of workload $1/\varepsilon$ released at time $\lceil \varepsilon D \rceil$. Note that until time $\lceil \varepsilon D \rceil$ the algorithm cannot tell the difference between instances $J_1$ and $J_2$.

Depending on how much the algorithm works before time $\lceil \varepsilon D \rceil$, we distinguish the following cases.

1. If the algorithm works more that $1/2$ then the energy spent by the algorithm until time $\lceil \varepsilon D \rceil$ is at least
$$(1/2)^\alpha/(\lceil \varepsilon D \rceil)^{\alpha-1} = \Omega\left(\frac{1}{\varepsilon D}\right)^{\alpha-1}.$$

2. However, if it works less than $1/2$ then on instance $J_2$, a total work of at least $(1/\varepsilon + 1 - 1/2) = (1/2 + 1/\varepsilon)$ remains to be done in $D$ time units. Hence the energy consumption on instance $J_2$ is at least
$$\frac{(1/2 + 1/\varepsilon)^\alpha}{D^{\alpha-1}}.$$

If the algorithm is $(1 + \varepsilon/3)^{\alpha-1}$-consistent, then it must be that the algorithm works more that $1/2$ before time $\lceil \varepsilon D \rceil$ otherwise, by the second case of the analysis, the competitive ratio is at least

$$\frac{(1/2 + 1/\varepsilon)^\alpha}{(1/\varepsilon)^\alpha(1 + \varepsilon)} = \frac{(1 + \varepsilon/2)^\alpha}{1 + \varepsilon} > (1 + \varepsilon/3)^{\alpha-1},$$

where the last inequality holds for $\alpha > 4$ and $\varepsilon$ small enough.

However it means that if the algorithm was running on instance $J_1$ (i.e. the prediction is incorrect) then by the first case the approximation ratio is at least $\Omega\left(\frac{1}{\varepsilon}\right)^{\alpha-1}$. $\qquad\square$

We then argue that one cannot hope to rely on some $l_p$ norm for $p < \alpha$ to measure error.

**Theorem 13.** *Fix some $\alpha$ and $D$ and let $p$ such that $p < \alpha$. Suppose there is an algorithm which on some prediction $w^{\mathrm{pred}}$ computes a solution of value at most*

$$C \cdot \mathrm{OPT} + C' \cdot \|w - w^{\mathrm{pred}}\|_p^p.$$

*Here $C$ and $C'$ are constants that can be chosen as an arbitrary function of $\alpha$ and $D$.*

*Then it also exists an algorithm for the online problem (without predictions) which is $(C + \varepsilon)$-competitive for every $\varepsilon > 0$.*

In other words, predictions do not help, if we choose $p < \alpha$.

*Proof.* In the online algorithm we use the prediction-based algorithm $A_P$ as a black box. We set the prediction $\tilde{w}$ to all $0$. We forward each job to $A_P$, but scale its work by a large factor $M$. It is obvious that by scaling the optimum of the instance increases exactly by a factor $M^\alpha$. The error in the prediction, however, increases less:

$$\|M \cdot w - M \cdot w^{\mathrm{pred}}\|_p^p = M^p \cdot \|w - w^{\mathrm{pred}}\|_p^p.$$

We run the jobs as $A_P$ does, but scale them down by $M$ again. Thus, we get a schedule of value

$$M^{-\alpha}(M^\alpha \cdot C \cdot \mathrm{OPT} + M^p \cdot C' \cdot \|w - w^{\mathrm{pred}}\|_p^p) = C \cdot \mathrm{OPT} + M^{p-\alpha} \cdot C' \cdot \|w - w^{\mathrm{pred}}\|_p^p. \quad (3)$$

Now if we choose $M$ large enough, the second term in (3) becomes insignificant. First, we relate the prediction error to the optimum. First note that

$$\mathrm{OPT} \geqslant (1/D^\alpha) \cdot \|w\|_\alpha^\alpha$$

since the optimum solution cannot be less expensive than running all jobs $i$ disjointly at speed $w_i/D$ for time $D$. Second note that $\|w\|_p^p \leqslant \|w\|_\alpha^\alpha$ since $|x|^p \leqslant |x|^\alpha$ for any $x \geqslant 1$ (recall that we assumed our workloads to be integral). Hence we get that,

$$\|w - w^{\mathrm{pred}}\|_p^p = \|w\|_p^p \leqslant D^\alpha \cdot \mathrm{OPT}.$$

Choosing $M$ sufficiently large gives $M^{p-\alpha}C'D^\alpha < \varepsilon$, which implies that (3) is at most $(C + \varepsilon)\mathrm{OPT}$. $\qquad\square$

# D   Extension to evolving predictors

In this section, we extend the result of Section 3 to the case where the algorithm is provided several predictions over time. In particular, we assume that the algorithm is provided a new prediction at each integral time $t$. The setting is natural as for a very long timeline, it is intuitive that the predictor might renew its prediction over time. Since making a mistake in the prediction of a very far future seems also less hurtful than making a mistake in predicting an immediate future, we define a generalized error metric incorporating this idea.

Let $0 < \lambda < 1$ be a parameter that describes how fast the confidence in a prediction deteriorates with the time until the expected arrival of the predicted job. Define the prediction received at time $t$ as a workload vector $w^{\mathrm{pred}}(t)$. Recall we are still considering the uniform deadlines case hence an instance is defined as a triplet $(w, D, T)$.

We then define the total error of a series of predictions as

$$\mathrm{err}^{(\lambda)} = \sum_t \sum_{i=t+1}^\infty |w_i^{\mathrm{real}} - w_i^{\mathrm{pred}}(t)|^\alpha \cdot \lambda^{i-t}.$$

In the following we reduce the evolving predictions model to the single prediction one.

We would like to prove similar results as in the single prediction setting with respect to $\mathrm{err}^{(\lambda)}$. In order to do so, we will split the instance into parts of bounded time horizon, solve each one independently with a single prediction, and show that this also gives a guarantee based on $\mathrm{err}^{(\lambda)}$. In particular, we will use the algorithm for the single prediction model as a black box.

The basic idea is as follows. If no job were to arrive for a duration of $D$, then the instance before this interval and afterwards can be solved independently. This is because any job in the earlier instance must finish before any job in the later instance can start. Hence, they cannot interfere. At random points, we ignore all jobs for a duration of $D$, thereby split the instance. The ignored jobs will be scheduled sub-optimally using AVR. If we only do this occasionally, i.e., after every intervals of length $\gg D$, the error we introduce is negligible.

We proceed by defining the splitting procedure formally. Consider the timeline as infinite in both directions. To split the instance, we define some interval length $2kD$, where $k \in \mathbb{N}$ will be specified later. We split the infinite timeline into contiguous intervals of length $2kD$. Moreover, we choose

an offset $x \in \{0, \cdots, k-1\}$ uniformly at random. Using these values, we define intervals $I_i = [2((i-1)k-x)D, 2(ik-x)D)$. We will denote by $t_i = (2(i-1)k-x)D$ the start time of the interval $I_i$. Consequently, the end of $I_i$ is $t_{i+1}$.

In each interval $I_i$, we solve the instance given by the jobs entirely contained in this interval using our algorithm with the most recent prediction as of time $t_i$, i.e., $w^{\text{pred}}(t_i)$, and schedule the jobs accordingly. We write $s^{\text{ALG}(i)}$ for this schedule. For the jobs that are overlapping with two contiguous intervals we schedule them independently using the AVERAGE RATE heuristic. The schedule for the jobs overlapping with intervals $I_i$ and $I_{i+1}$ will be referred to as $s^{\text{AVR}(i)}$.

It is easy to see that this algorithm is robust: The energy of the produced schedule is

$$\int \left( \sum_i \left[ s^{\text{ALG}(i)}(t) + s^{\text{AVR}(i)}(t) \right] \right)^\alpha dt$$

$$\leqslant 2^\alpha \int \left( \sum_i s^{\text{ALG}(i)}(t) \right)^\alpha dt + 2^\alpha \int \left( \sum_i s^{\text{AVR}(i)}(t) \right)^\alpha dt.$$

Moreover, the first term can be bounded by $2^\alpha \cdot O(\alpha/\varepsilon)^\alpha$ OPT using Theorem 8 and the second term can be bounded by $2^\alpha \cdot 2^\alpha$ OPT because of Theorem 10. This gives an overall bound of $O(\alpha/\varepsilon)^\alpha$ on the competitive ratio.

In the rest of the section we focus on the consistency/smoothness guarantee. We first bound the costs of $s^{\text{ALG}(i)}$ and $s^{\text{AVR}(i)}$ isolated (ignoring potential interferences). Using these bounds, we derive an overall guarantee for the algorithm's cost.

**Lemma 14.**

$$\mathbb{E}\left( \sum_i \int s^{\text{AVR}(i)}(t)^\alpha \right) \leqslant \frac{2^\alpha}{k} \text{ OPT}$$

*Proof.* Fix some $i > 0$ and let us call $O_i$ the job instance consisting of jobs overlapping with both intervals $I_i$ and $I_{i+1}$. By Theorem 10 the energy used by AVR is at most a $2^\alpha$-factor from the optimum schedule. Hence,

$$\int s^{\text{AVR}(i)}(t)^\alpha dt \leqslant 2^\alpha \text{ OPT}(O_i).$$

Now denote by $s^{\text{OPT}}$ the speed function of the optimum schedule over the whole instance. Then

$$\text{OPT}(O_i) \leqslant \int_{t_i-D}^{t_i+D} s^{\text{OPT}}(t)^\alpha dt.$$

This holds because $s^{\text{OPT}}$ processes some work during $[t_i - D, t_i + D]$ which has to include all of $O_i$. Hence, we have that

$$\mathbb{E}\left( \sum_i \text{OPT}(O_i) \right)$$

$$\leqslant \frac{1}{k} \sum_{x=0}^{k-1} \sum_i \int_{2(ik-x)D-D}^{2(ik-x)D+D} s^{\text{OPT}}(t)^\alpha dt$$

$$\leqslant \frac{1}{k} \int s^{\text{OPT}}(t)^\alpha dt = \frac{1}{k} \text{ OPT}$$

The second inequality holds, because the integrals are over disjoint ranges. Together, with the bound on $s^{\text{AVR}(i)}$ we get the claimed inequality. $\square$

**Lemma 15.**

$$\sum_i \int s^{\text{ALG}(i)}(t)^\alpha dt \leqslant (1+\varepsilon) \text{ OPT} + O\left(\frac{\alpha}{\varepsilon}\right)^\alpha \cdot \lambda^{-2kD} \cdot \text{err}^{(\lambda)}.$$

*Proof.* Note that for any $i$

$$\sum_{t=(t_i)+1}^{t_{(i+1)}} |w_t^{\text{real}} - w_t^{\text{pred}}(t_i)|^\alpha \leqslant \lambda^{-2kD} \sum_{t=(t_i)+1}^{t_{(i+1)}} |w_t^{\text{real}} - w_t^{\text{pred}}(t_i)|^\alpha \lambda^{t-t_i}.$$

Hence,

$$\sum_i \sum_{t=t_i}^{t_{i+1}} |w_t^{\text{real}} - w_t^{\text{pred}}(t_i)|^\alpha \leqslant \lambda^{-2kD} \operatorname{err}^{(\lambda)}.$$

Using Theorem 8 for each $\int s_{\text{ALG}}^{(i)}(t)^\alpha dt$, we get a bound depending on $\sum_{t=t_i}^{t_{i+1}} |w_t^{\text{real}} - w_t^{\text{pred}}(t_i)|^\alpha$. Summing over $i$ and using the inequality above finishes the proof of the lemma. $\square$

We are ready to state the consistency/smoothness guarantee of the splitting algorithm.

**Theorem 16.** *With robustness parameter $O(\varepsilon/\alpha)$ the splitting algorithm produces in expectation a schedule of cost at most*

$$(1+\varepsilon)\operatorname{OPT} + O\left(\frac{\alpha}{\varepsilon}\right)^\alpha \cdot \lambda^{-D/\varepsilon \cdot O(\alpha/\varepsilon)^\alpha} \cdot \operatorname{err}^{(\lambda)}.$$

In other words, we get the same guarantee as in the single prediction case, except that the dependency on the error is larger by a factor of $\lambda^{-D/\varepsilon \cdot O(\alpha/\varepsilon)^\alpha}$. The exponential dependency on $D$ may seem unsatisfying, but (1) it cannot be avoided (see Theorem 17) and (2) for moderate values of $\lambda$, e.g. $\lambda = 1 - 1/D$, this exponential dependency vanishes.

*Proof.* We will make use of the following inequality: For all $a, b \geqslant 0$ and $0 < \delta \leqslant 1$, it holds that

$$(a+b)^\alpha \leqslant (1+\delta)a^\alpha + \left(\frac{3\alpha}{\delta}\right)^\alpha b^\alpha.$$

This follows from a simple case distinction whether $b \leqslant a \cdot \delta/(2\alpha)$. In expectation, the cost of the algorithm is bounded by

$$\mathbb{E}\left[\int\left(\sum_i [s^{\text{ALG}(i)}(t) + s^{\text{AVR}(i)}(t)]\right)^\alpha dt\right]$$

$$\leqslant (1+\varepsilon)\mathbb{E}\left[\int\sum_i (s^{\text{ALG}(i)}(t))^\alpha dt\right]$$

$$+ \left(\frac{3\alpha}{\varepsilon}\right)^\alpha \mathbb{E}\left[\int\sum_i (s^{\text{AVR}(i)}(t))^\alpha dt\right]$$

$$\leqslant (1+\varepsilon)\int\sum_i s^{\text{ALG}(i)}(t))^\alpha dt$$

$$+ \frac{1}{k}\left(\frac{6\alpha}{\varepsilon}\right)^\alpha \operatorname{OPT}.$$

By choosing $k = 1/\varepsilon(6\alpha/\varepsilon)^\alpha$ the latter term becomes $\varepsilon\operatorname{OPT}$. With Lemma 15 we can bound the term above by

$$(1+\varepsilon)^3\operatorname{OPT} + O\left(\frac{\alpha}{\varepsilon}\right)^\alpha \cdot \lambda^{-D/\varepsilon \cdot O(\alpha/\varepsilon)^\alpha} \cdot \operatorname{err}^{(\lambda)}.$$

Scaling $\varepsilon$ by a constant yields the claimed guarantee. $\square$

We complement the result of this section with an impossibility result. We allow the parameter $\lambda$ in the definition of $\operatorname{err}^{(\lambda)}$ to be a function of $D$ and we write $\lambda(D)$.

**Theorem 17.** *Let $\operatorname{err}(\lambda)$ the error in the evolving prediction model be defined with some $0 < \lambda(D) < 1$ that can depend on $D$. Suppose there is an algorithm which computes a solution of value at most*

$$C \cdot \operatorname{OPT} + C'(D) \cdot \operatorname{err}^{(\lambda)},$$

*where $C$ is independent of $D$ and $C'(D) = o\left(\frac{1-\lambda(D)^D}{\lambda(D)^D} \cdot \frac{1}{D^\alpha}\right)$. Then there also exists an algorithm for the online problem (without predictions) which is $(C+\varepsilon)$-competitive for every $\varepsilon > 0$.*

In particular, note that for $\lambda$ independent of $D$, it shows that an exponential dependency in $D$ is needed in $C'(D)$ as we get in Theorem 16.

*Proof.* The structure of the proof is similar to that of Theorem 13. We pass an instance to the assumed algorithm, but set the prediction to all $0$. Unlike the previous proof, we keep the same workloads when passing the jobs, but subdivide $D$ in to $D \cdot k$ time steps where $k$ will be specified later. This will decrease the cost of every solution by $k^\alpha$.

Take an instance with interval length $D$. Like in the proof of Theorem 13 we have that

$$\|w^{\mathrm{real}}\|_\alpha^\alpha \leqslant D^\alpha \cdot \mathrm{OPT}.$$

Consider the error parameter $\mathrm{err}^{(\lambda)\prime}$ for the instance with $D' = D \cdot k$. We observe that

$$\mathrm{err}^{(\lambda)\prime} = \sum_t \sum_{i=t+1}^{\infty} |w_{k \cdot i}^{\mathrm{real}}|^\alpha \cdot \lambda(D')^{k(i-t)}$$

$$\leqslant \|w^{\mathrm{real}}\|_\alpha^\alpha \cdot \sum_{i=1}^{\infty} \lambda(D')^{k \cdot i}$$

$$\leqslant \|w^{\mathrm{real}}\|_\alpha^\alpha \frac{\lambda(D')^k}{1 - \lambda(D')^k}$$

$$\leqslant D^\alpha \frac{\lambda(D')^k}{1 - \lambda(D')^k} \cdot \mathrm{OPT}$$

Hence, by definition the algorithm produces a solution of cost

$$C \cdot \mathrm{OPT} / k^\alpha + C'(D') \, \mathrm{err}^{(\lambda)\prime} \leqslant (C/k^\alpha + D^\alpha \frac{\lambda(D')^k}{1 - \lambda(D')^k} C'(D')) \cdot \mathrm{OPT}$$

for the subdivided instance. Transferring it to the original instance, we get a cost of

$$(C + k^\alpha D^\alpha \frac{\lambda(D')^k}{1 - \lambda(D')^k} C'(D')) \cdot \mathrm{OPT}$$

Therefore, if $k^\alpha \frac{\lambda(D \cdot k)^k}{1 - \lambda(D \cdot k)^k} C'(D \cdot k)$ tends to $0$ as $k$ grows, for any $\varepsilon > 0$, we can fix $k$ big enough so that the cost of the algorithm is at most $(C + \varepsilon) \, \mathrm{OPT}$. $\square$

# E   A shrinking lemma

Recall that by applying the earliest-deadline-first policy, we can normalize every schedule to run at most one job at each time. We say, it is run *isolated*. Moreover, if a job is run isolated, it is always better to run it at a uniform speed (by convexity of $x \mapsto x^\alpha$ on $x \geqslant 0$). Hence, an optimal schedule can be characterized solely by the total time $p_j$ each job is run. Given such $p_j$ we will give a necessary and sufficient condition of when a schedule that runs each job isolated for $p_j$ time exists. Note that we assume we are in the general deadline case, each job $j$ comes with a release $r_j$ and deadline $d_j$ and the EDF policy might cause some jobs to be preempted.

**Lemma 18.** *Let there be a set of $n$ jobs with release times $r_j$ and deadlines $d_j$ for each job $j$. Let $p_j$ denote the total duration that $j$ should be processed. Scheduling the jobs isolated earliest-deadline-first, with the constraint to never run a job before its release time, will complete every job $j$ before time $d_j$ if and only if for every interval $[t, t']$ it holds that*

$$\sum_{j : t \leqslant r_j, d_j \leqslant t'} p_j \leqslant t' - t \tag{4}$$

*Proof.* For the one direction, let $t, t'$ such that (4) is not fulfilled. Since the jobs with $t \leqslant r_j$ cannot be processed before $t$, the last such job $j'$ to be completed must finish after

$$t + \sum_{j : t \leqslant r_j, d_j \leqslant t'} p_j > t + t' - t = t' \geqslant d_{j'}$$

For the other direction, we will schedule the jobs earliest-deadline-first and argue that if the schedule completes some job after its deadline, then (4) is not satisfied for some interval $[t, t']$.

To this end, let $j'$ be the first job that finishes strictly after $d_{j'}$ and consider the interval $I_0 = [r_{j'}, d_{j'}]$. We now define the following operator that transforms our interval $I_0$ into an interval $I_1$. Consider $t_{\text{inf}}$ to be the smallest release time among all jobs that are processed in interval $I_0$ and define $I_1 = [t_{\text{inf}}, d_{j'}]$. We apply iteratively this operation to obtain interval $I_{k+1}$ from interval $I_k$. We claim the following properties that we prove by induction.

1. For any $k \geqslant 0$, the machine is never idle in interval $I_k$.

2. For any $k \geqslant 0$, all jobs that are processed in $I_k$ have a deadline $\leqslant d_{j'}$.

For $I_0 = [r_{j'}, d_{j'}]$, since job $j'$ is not finished by time $d_{j'}$ it must be that the machine is never idle in that interval. Additionally, if a job is processed in this interval, it must be that its deadline is earlier that $d_{j'}$ since we process in EDF order. Assume both items hold for $I_k$ and then consider $I_{k+1}$ that we denote by $[a_{k+1}, d_{j'}]$. By construction, there is a job denoted $j_{k+1}$ released at time $a_{k+1}$ that is not finished by time $a_k$. Therefore the machine cannot be idle at any time in $[a_{k+1}, a_k]$ hence at any time in $I_{k+1}$ by the induction hypothesis. Furthermore, consider a job processed in $I_{k+1} \setminus I_k$. It must be that its deadline is earlier that the deadline of job $j_{k+1}$. But job $j_{k+1}$ is processed in interval $I_k$ which implies that its deadline is earlier than $d_{j'}$ and ends the induction.

Denote by $k'$ the first index such that $I_{k'} = I_{k'+1}$. We define $I_\infty = I_{k'}$. By construction, it must be that all jobs processed in $I_\infty$ have release time in $I_\infty$ and by induction the machine is never idle in this interval and all jobs processed in $I_\infty$ have deadline in $I_\infty$.

Since job $j'$ is not finished by time $d_{j'}$ and by the previous remarks we have that

$$\sum_{j:r_j,d_j \in I_\infty} p_j > |I_\infty|$$

which yields a counter example to (4). □

We can now prove two shrinking lemmas that are needed in the procedure ROBUSTIFY and its generalization to general deadlines.

**Lemma 19.** *Let $0 \leqslant \mu < 1$. For any instance $\mathcal{I}$ consider the instance $\mathcal{I}'$ where the deadline of job $j$ is set to $d'_j = r_j + (1 - \mu)(d_j - r_j)$ (i.e. we shrink each job by a $(1 - \mu)$ factor). Then*

$$\text{OPT}(\mathcal{I}') \leqslant \frac{\text{OPT}(\mathcal{I})}{(1 - \mu)^{\alpha - 1}}$$

*Additionally, assuming $0 \leqslant \mu < 1/2$, consider the instance $\mathcal{I}''$ where the deadline of job $j$ is set to $d''_j = r_j + (1 - \mu)(d_j - r_j)$ and the release time is set to $r''_j = r_j + \mu(d_j - r_j)$. Then*

$$\text{OPT}(\mathcal{I}'') \leqslant \frac{\text{OPT}(\mathcal{I})}{(1 - 2\mu)^{\alpha - 1}}$$

*Proof.* W.l.o.g. we can assume that the optimal schedule $s$ for $\mathcal{I}$ runs each job isolated and at a uniform speed. By optimality of the schedule and convexity, each job $j$ must be run at a constant speed $s_j$ for a total duration of $p_j$. Consider the first case and define a speed $s'_j = \frac{s_j}{1-\mu}$ for all $j$ (hence the total processing time becomes $p'_j = (1 - \mu) \cdot p_j$).

Assume now in the new instance $\mathcal{I}'$ we run jobs earliest-deadline-first with the constraint that no job is run before its release time (with the processing times $p'_j$). We will prove using Lemma 18 that all deadlines are satisfied. Consider now an interval $[t, t']$ we then have that

$$\sum_{j:t \leqslant r_j, d'_j \leqslant t'} p'_j = (1 - \mu) \cdot \sum_{j:t \leqslant r_j, d'_j \leqslant t'} p_j \leqslant (1 - \mu) \cdot \sum_{j:t \leqslant r_j, d_j \leqslant \frac{t' - \mu t}{1 - \mu}} p_j$$

where the last inequality comes from the fact that $t' \geqslant d'_j = d_j - \mu(d_j - r_j)$ which implies that $d_j \leqslant \frac{t' - \mu r_j}{1 - \mu} \leqslant \frac{t' - \mu t}{1 - \mu}$ by using $r_j \geqslant t$. By Lemma 18 and the fact that $s$ is a feasible schedule for $\mathcal{I}$

we have that

$$\sum_{j:t\leqslant r_j,d_j'\leqslant t'} p_j' \leqslant (1-\mu)\cdot\left(\frac{t'-\mu t}{1-\mu}-t\right) = (1-\mu)\cdot\frac{t'-t}{1-\mu} = t'-t$$

which implies by Lemma 18 that running all jobs EDF with processing time $p_j'$ satisfies all deadlines $d_j'$. Now notice the cost of this schedule is at most $\frac{1}{(1-\mu)^{\alpha-1}}$ times the original schedule $s$ which ends the proof (each job is ran $\frac{1}{1-\mu}$ times faster but for a time $(1-\mu)$ times shorter).

The proof of the second case is similar. Note that for any $[t,t']$, if

$$d_j'' = r_j + (1-\mu)(d_j - r_j) = (1-\mu)d_j + \mu r_j \leqslant t'$$
$$r_j'' = r_j + \mu(d_j - r_j) = (1-\mu)r_j + \mu d_j \geqslant t$$

then we have

$$(1-\mu)d_j \leqslant t' - \mu r_j \leqslant t' - \frac{\mu}{1-\mu}(t-\mu d_j)$$

$$\iff (1-\mu)d_j - \frac{\mu^2}{1-\mu}d_j \leqslant t' - \frac{\mu}{1-\mu}\cdot t$$

$$\iff d_j((1-\mu)^2 - \mu^2) \leqslant (1-\mu)t' - \mu t$$

$$\iff d_j \leqslant \frac{(1-\mu)t' - \mu t}{1-2\mu}$$

Similarly, we have

$$(1-\mu)r_j \geqslant t - \mu d_j \geqslant t - \frac{\mu}{1-\mu}(t' - \mu r_j)$$

$$\iff (1-\mu)r_j - \frac{\mu^2}{1-\mu}r_j \geqslant t - \frac{\mu}{1-\mu}\cdot t'$$

$$\iff r_j \geqslant \frac{(1-\mu)t - \mu t'}{1-2\mu}$$

Notice that $\frac{(1-\mu)t'-\mu t}{1-2\mu} - \frac{(1-\mu)t-\mu t'}{1-2\mu} = \frac{t'-t}{1-2\mu}$

Therefore, if we set the speed that each job $s_j''$ is processed to $s_j'' = \frac{s_j}{1-2\mu}$ then we have a processing time $p_j'' = (1-2\mu)\cdot p_j$ and we can write

$$\sum_{j:t\leqslant r_j'',d_j''\leqslant t'} p_j'' = (1-2\mu)\cdot\sum_{j:t\leqslant r_j'',d_j''\leqslant t'} p_j$$

$$\leqslant (1-2\mu)\cdot\sum_{j:\frac{(1-\mu)t-\mu t'}{1-2\mu}\leqslant r_j,d_j\leqslant\frac{(1-\mu)t'-\mu t}{1-2\mu}} p_j$$

$$\leqslant (1-2\mu)\cdot\frac{t'-t}{1-2\mu} = t'-t$$

by Lemma 18. Hence we can conclude similarly as in the previous case. $\qquad\square$

# F   Making an algorithm noise tolerant

The idea for achieving noise tolerance is that by Lemma 19 we know that if we delay each job's arrival slightly (e.g., by $\eta D$) we can still obtain a near optimal solution. This gives us time to reassign arriving jobs within a small interval in order to make the input more similar to the prediction. We first, in Section F.1, generalize the error function err to a more noise tolerant error function $\text{err}_\eta$. We then, in Section F.2, give a general procedure for making an algorithm noise tolerant (see Theorem 20).

### F.1 Noise tolerant measure of error

For motivation, recall the example given in the main body. Specifically, consider a predicted workload $w^{\mathrm{pred}}$ defined by $w_i^{\mathrm{pred}} = 1$ for those time steps $i$ that are divisible by a large constant, say 1000, and let $w_i^{\mathrm{pred}} = 0$ for all other time steps. If the real instance $w^{\mathrm{real}}$ is a small shift of $w^{\mathrm{pred}}$ say $w_{i+1}^{\mathrm{real}} = w_i^{\mathrm{pred}}$ then the prediction error $\mathrm{err}(w^{\mathrm{real}}, w^{\mathrm{pred}})$ is large although $w^{\mathrm{pred}}$ intuitively forms a good prediction of $w^{\mathrm{real}}$. To overcome this sensitivity to noise, we generalize the definition of err.

For two workload vectors $w, w'$, and a parameter $\eta \geqslant 0$, we say that $w$ is in the $\eta$-neighborhood of $w'$, denoted by $w \in N_\eta(w')$, if $w$ can be obtained from $w'$ by moving the workload at most $\eta D$ time steps forward or backward in time. Formally $w \in N(w')$ if there exists a solution $\{x_{ij}\}$ to the following system of linear equations[3]:

$$w_i = \sum_{j=i-\eta D}^{i+\eta D} x_{ij} \qquad \forall i$$

$$w'_j = \sum_{i=j-\eta D}^{j+\eta D} x_{ij} \qquad \forall j$$

The concept of $\eta$-neighborhood is inspired by the notion of earth mover's distance but is adapted to our setting. Intuitively, the variable $x_{ij}$ denotes how much of the load $w_i$ has been moved to time unit $j$ in order to obtain $w'$. Also note that it is a symmetric and reflexive relation, i.e., if $w \in N_\eta(w')$ then $w' \in N_\eta(w)$ and $w \in N_\eta(w)$.

We now generalize the measure of prediction error as follows. For a parameter $\eta \geqslant 0$, an instance $w^{\mathrm{real}}$, and a prediction $w^{\mathrm{pred}}$, we define the $\eta$-prediction error, denoted by $\mathrm{err}_\eta$, as

$$\mathrm{err}_\eta(w^{\mathrm{real}}, w^{\mathrm{pred}}) = \min_{w \in N_\eta(w^{\mathrm{pred}})} \mathrm{err}(w^{\mathrm{real}}, w).$$

Note that by symmetry we have that $\mathrm{err}_\eta(w^{\mathrm{real}}, w^{\mathrm{pred}}) = \mathrm{err}_\eta(w^{\mathrm{pred}}, w^{\mathrm{real}})$. Furthermore, we have that $\mathrm{err}_\eta = \mathrm{err}$ if $\eta = 0$ but it may be much smaller for $\eta > 0$. To see this, consider the vectors $w^{\mathrm{pred}}$ and $w_i^{\mathrm{real}} = w_{i+1}^{\mathrm{pred}}$ given in the motivational example above. While $\mathrm{err}(w^{\mathrm{pred}}, w^{\mathrm{real}})$ is large, we have $\mathrm{err}_\eta(w^{\mathrm{pred}}, w^{\mathrm{real}}) = 0$ for any $\eta$ with $\eta D \geqslant 1$. Indeed the definition of $\mathrm{err}_\eta$ is exactly so as to allow for a certain amount of noise (calibrated by the parameter $\eta$) in the prediction.

### F.2 Noise tolerant procedure

We give a general procedure for making an algorithm $\mathcal{A}$ noise tolerant under the mild condition that $\mathcal{A}$ is monotone: we say that an algorithm is monotone if given a predictor $w^{\mathrm{pred}}$ and duration $D$, the cost of scheduling a workload $w$ is at least as large as that of scheduling a workload $w'$ if $w \geqslant w'$ (coordinate-wise). That increasing the workload should only increase the cost of a schedule is a natural condition that in particular all our algorithms satisfy.

**Theorem 20.** *Suppose there is a monotone learning-augmented online algorithm $\mathcal{A}$ for the uniform speed scaling problem, that given prediction $w^{\mathrm{pred}}$, computes a schedule of an instance $w^{\mathrm{real}}$ of value at most*

$$\min\{C \cdot \mathrm{OPT} + C' \, \mathrm{err}(w^{\mathrm{real}}, w^{\mathrm{pred}}), C'' \, \mathrm{OPT}\}.$$

*Then, for every $\eta \geqslant 0$, $\zeta > 0$ there is a learning-augmented online algorithm* NOISE-ROBUST$(\mathcal{A})$, *that given prediction $w^{\mathrm{pred}}$, computes a schedule of $w^{\mathrm{real}}$ of value at most $((1+\eta)(1+\zeta))^{O(\alpha)}$ times*

$$\min\{C \cdot \mathrm{OPT} + (1/\zeta)^{O(\alpha)}(C + C') \, \mathrm{err}_\eta(w^{\mathrm{real}}, w^{\mathrm{pred}}), C'' \, \mathrm{OPT}\}.$$

The pseudo-code of the online algorithm NOISE-ROBUST$(\mathcal{A})$, obtained from $\mathcal{A}$, is given in Algorithm 2.

**Algorithm 2** NOISE-ROBUST($\mathcal{A}$)

---

**Input:** Algorithm $\mathcal{A}$, prediction $w^{\text{pred}}$, and $\eta \geqslant 0, \zeta > 0$

1: Initialize $\mathcal{A}$ with prediction $\overline{w}_i^{\text{pred}} = (1+\zeta)w_{i-\eta D}^{\text{pred}}$ and duration $(1-2\eta)D$
2: Let $w^{\text{online}}$ and $\overline{w}^{\text{real}}$ be workload vectors, initialized to 0
3: **on time step $i$ do**
4:     $W \leftarrow w_i^{\text{real}}$
5:     **for** $j \in \{i - \eta D, \dots, i + \eta D\}$ **do**
6:         **if** $w_j^{\text{online}} + W \leqslant (1+\zeta)w_j^{\text{pred}}$ **then**
7:             $x_{ij} \leftarrow W$
8:             $W \leftarrow 0$
9:             $w_j^{\text{online}} \leftarrow w_j^{\text{online}} + W$
10:        **else if** $w_j^{\text{online}} < (1+\zeta)w_j^{\text{pred}}$ **then**
11:           $x_{ij} \leftarrow (1+\zeta)w_j^{\text{pred}} - w_j^{\text{online}}$
12:           $W \leftarrow W - x_{ij}$
13:           $w_j^{\text{online}} \leftarrow (1+\zeta)w_j^{\text{pred}}$
14:        **end if**
15:     **end for**
16:     *// Distribute remaining workload $W$ evenly*
17:     **for** $j \in \{i - \eta D, \dots, i + \eta D\}$ **do**
18:         $x_{ij} \leftarrow x_{ij} + W/(2\eta D + 1)$
19:         $w_j^{\text{online}} \leftarrow w_j^{\text{online}} + W/(2\eta D + 1)$
20:     **end for**
21:     $\overline{w}_i^{\text{real}} \leftarrow w_{i-\eta D}^{\text{online}}$
22:     Feed the job with workload $\overline{w}_i^{\text{real}}$ to $\mathcal{A}$
23: **end on**

---

Figure 3: An example of the construction of the vector $w^{\text{online}}$ from $w^{\text{real}}$ and $w^{\text{pred}}$.

The algorithm constructs a vector $w^{\text{online}} \in N_\eta(w^{\text{real}})$ while trying to minimize $\text{err}(w^{\text{online}}, w^{\text{pred}})$. Each component $w_i^{\text{online}}$ will be finalized at time $i + \eta D$. Hence, we forward the jobs to $\mathcal{A}$ with a delay of $\eta D$.

The vector is constructed as follows. Suppose a job $w_i^{\text{real}}$ arrives. The algorithm first (see Steps 4-15) greedily assigns the workload to the time steps $j = i - \eta D, i - \eta D + 1, \dots, i + \eta D$ from left-to-right subject to the constraint that no time step receives a workload higher than $(1+\zeta)w_j^{\text{pred}}$. If not all workload of $w_i^{\text{real}}$ was assigned in this way, then the overflow is assigned uniformly to the time steps from $i - \eta D$ to $i + \eta D$ (Steps 17-20). Since each $w_j^{\text{online}}$ can only receive workloads during time steps $j - \eta D, \dots, j + \eta D$, it will be finalized at time $j + \eta D$. Thus, at time $i$ we can safely forward $w_{i-\eta D}^{\text{online}}$ to the algorithm $\mathcal{A}$. Hence, we set the workload of the algorithm's instance to $\overline{w}_i^{\text{real}} = w_{i-\eta D}^{\text{online}}$ (Steps 21-22). This shift together with the fact that a job $w_i^{\text{real}}$ may be assigned to $w_{i+\eta D}^{\text{online}}$, i.e., $\eta D$ time steps forward in time, is the reason why we run each job with an interval of length $(1-2\eta)D$. Shrinking the interval of each job allows to make this shift and reassignment while still guaranteeing that each job is finished by its original deadline.

For an example, consider Figure 3. Here we assume that $\eta D = 1$ and for illustrative purposes that $\zeta = 0$. At time 0, a workload $w_0^{\text{real}} = 1$ is released. The algorithm NOISE-ROBUST$(\mathcal{A})$ then greedily constructs $w^{\text{online}}$ by filling the available slots in $w_{-1}^{\text{pred}}, w_0^{\text{pred}}$, and $w_1^{\text{pred}}$. Since $w_0^{\text{pred}} = 3$, it fits all of the workload of $w_0^{\text{real}}$ at time 0. Similarly the workloads $w_2^{\text{real}}$ and $w_3^{\text{real}}$ both fit under the capacity given by $w^{\text{pred}}$. Now consider the workload $w_4^{\text{real}} = 2$ released at time 4. At this point, the available workload at time 2 is fully occupied and one there is one unit of workload left at time 3. Hence, NOISE-ROBUST$(\mathcal{A})$ will first assign the one unit of $w_4^{\text{real}}$ to the third time slot and then split the remaining unit of workload unit uniformly across the time steps $3, 4, 5$. The obtained vector $w^{\text{online}}$ is depicted on the right of Figure 3. The workload $w^{\text{online}}$ is then fed online to the algorithm $\mathcal{A}$ (giving a schedule of $w^{\text{online}}$ and thus of $w^{\text{real}}$) so that at time $i$, $\mathcal{A}$ receives the job $\overline{w}_i^{\text{real}} = w_{i+\eta D}^{\text{online}} = w_{i+1}^{\text{online}}$ with a deadline of $i + (1 - 2\eta)D = i + D - 2$. This deadline is chosen so as to guarantee that a job is finished by $\mathcal{A}$ within its original deadline. Indeed, by this selection, the last part of the job $w_4^{\text{real}}$ that was assigned to $w_5^{\text{online}}$ is guaranteed to finish by time $6 + D - 2 = 4 + D$ which is its original deadline.

Having described the algorithm, we proceed to analyze its guarantees which will prove Theorem 20.

**Analysis.** We start by analyzing the noise tolerance of NOISE-ROBUST$(\mathcal{A})$.

**Lemma 21.** *The schedule computed by* NOISE-ROBUST$(\mathcal{A})$ *has cost at most* $(1 + O(\eta))^\alpha C''$ OPT.

*Proof.* Let OPT and OPT$'$ denote the cost of an optimum schedule of the original instance $w^{\text{real}}$ with duration $D$ and the instance $\overline{w}^{\text{real}}$ with duration $(1 - 2\eta)D$ fed to $\mathcal{A}$, respectively. The lemma then follows by showing that

$$\text{OPT}' \leqslant (1 + O(\eta))^\alpha \text{ OPT} .$$

To show this inequality, consider an optimal schedule $s$ of $w^{\text{real}}$ subject to the constraint that every job $w_i^{\text{real}}$ is scheduled within the time interval $[i + 2\eta D, i + (1 - 2\eta)D]$. By Lemma 19, we have that the cost of this schedule is at most $(1 + O(\eta))^\alpha$ OPT. The statement therefore follows by arguing that $s$ also gives a feasible schedule of $\overline{w}^{\text{real}}$ with duration $(1 - 2\eta)D$. To see this note that NOISE-ROBUST$(\mathcal{A})$ moves the workload $w_i^{\text{real}}$ to a subset of $\overline{w}_i^{\text{real}}, \overline{w}_{i+1}^{\text{real}}, \ldots, \overline{w}_{i+2\eta D}^{\text{real}}$. All of these jobs are allowed to be processed during $[i + 2\eta D, i + (1 - 2\eta)D]$. It follows that the part of these jobs that corresponds to $w_i^{\text{real}}$ can be processed in the computed schedule $s$ (whenever it processes $w_i^{\text{real}}$) since $s$ process that job in the time interval $[i + 2\eta D, i + (1 - 2\eta)D]$. By doing this "reverse-mapping" for every job, we can thus use $s$ as a schedule for the instance $\overline{w}^{\text{real}}$ with duration $(1 - 2\eta)D$. $\square$

We now proceed to analyze the consistency and smoothness. The following lemma is the main technical part of the analysis. We use the common notation $(a)^+$ for $\max\{a, 0\}$.

**Lemma 22.** *The workload vector $w^{\text{online}}$ produced by* NOISE-ROBUST$(\mathcal{A})$ *satisfies*

$$\sum_i \left[ \left( w_i^{\text{online}} - (1 + \zeta)w_i^{\text{pred}} \right)^+ \right]^\alpha \leqslant O(1/\zeta)^{3\alpha} \cdot \min_{w \in N_\eta(w^{\text{real}})} \sum_i \left[ \left( w_i - w_i^{\text{pred}} \right)^+ \right]^\alpha .$$

The more technical proof of this lemma is given in Section F.2.1. Here, we explain how it implies the consistency and smoothness bounds of Theorem 20. For a workload vector $w$, we use the notation $\text{OPT}(w)$ and $\text{OPT}'(w)$ to denote the cost of an optimal schedule of workload $w$ with duration $D$ and $(1 - 2\eta)D$, respectively. Now let $\hat{w}^{\text{online}}$ be the workload vector defined by

$$\hat{w}_i^{\text{online}} = \max\{w_i^{\text{online}}, (1 + \zeta)w_i^{\text{pred}}\} .$$

We analyze the cost of the schedule produced by $\mathcal{A}$ for $\hat{w}^{\text{online}}$ (shifted by $\eta D$). This also bounds the cost of running $\mathcal{A}$ with $\overline{w}^{\text{real}}$: Since $\mathcal{A}$ is monotone, the cost of the schedule computed for the workload $\hat{w}^{\text{online}}$ (shifted by $\eta D$) can only be greater than that computed for $\overline{w}^{\text{real}}$ which equals $w^{\text{online}}$ (shifted by $\eta D$). Furthermore, we have by Lemma 22 that

$$\text{err}(\hat{w}^{\text{online}}, (1 + \zeta)w^{\text{pred}}) = \sum_i \left[ \left( w_i^{\text{online}} - (1 + \zeta)w_i^{\text{pred}} \right)^+ \right]^\alpha \tag{5}$$

$$\leqslant O(1/\zeta)^{3\alpha} \text{err}_\eta(w^{\text{real}}, w^{\text{pred}}) .$$

It follows by the assumptions on $\mathcal{A}$ that the schedule computed by NOISE-ROBUST($\mathcal{A}$) has cost at most

$$C \cdot \mathrm{OPT}'(\hat{w}^{\mathrm{online}}) + C' \cdot \mathsf{err}(\hat{w}^{\mathrm{online}}, (1 + \zeta)w^{\mathrm{pred}})$$
$$\leqslant C \cdot \mathrm{OPT}'(\hat{w}^{\mathrm{online}}) + O(1/\zeta)^{3\alpha} \cdot C' \cdot \mathsf{err}_\eta(w^{\mathrm{real}}, w^{\mathrm{pred}}) \, .$$

The following lemma implies the consistency and smoothness, as stated in Theorem 20, by relating $\mathrm{OPT}'(\hat{w}^{\mathrm{online}})$ with the cost $\mathrm{OPT} = \mathrm{OPT}(w^{\mathrm{real}})$.

**Lemma 23.** *We have*

$$\mathrm{OPT}'(\hat{w}^{\mathrm{online}}) \leqslant ((1 + \eta)(1 + \zeta))^{O(\alpha)} \left( \mathrm{OPT}(w^{\mathrm{real}}) + O(1/\zeta)^{4\alpha} \, \mathsf{err}_\eta(w^{\mathrm{real}}, w^{\mathrm{pred}}) \right) \, .$$

*Proof.* By the exact same arguments as in the proof of Theorem 2, we have that for any $\eta' > 0$

$$\mathrm{OPT}'(\hat{w}^{\mathrm{online}}) \leqslant (1 + \eta')^\alpha \, \mathrm{OPT}'((1 + \zeta)w^{\mathrm{pred}}) + O(1/\eta')^\alpha \, \mathsf{err}(\hat{w}^{\mathrm{online}}, (1 + \zeta)w^{\mathrm{pred}})$$
$$\leqslant (1 + \eta')^\alpha \, \mathrm{OPT}'((1 + \zeta)w^{\mathrm{pred}}) + O(1/\eta')^\alpha O(1/\zeta)^{3\alpha} \, \mathsf{err}_\eta(w^{\mathrm{real}}, w^{\mathrm{pred}}) \, ,$$

where we used (5) for the second inequality.

By Lemma 19, we have that decreasing the duration by a factor $(1 - 2\eta)$ only increases the cost by factor $(1 + O(\eta))^\alpha$ and so $\mathrm{OPT}'((1 + \zeta)w^{\mathrm{pred}}) \leqslant (1 + O(\eta))^\alpha \, \mathrm{OPT}((1 + \zeta)w^{\mathrm{pred}})$. Furthermore, as a schedule for a workload $w^{\mathrm{pred}}$ gives a schedule for $(1 + \zeta)w^{\mathrm{pred}}$ by increasing the speed by a factor $(1 + \zeta)$, we get

$$\mathrm{OPT}'((1 + \zeta)w^{\mathrm{pred}}) \leqslant (1 + O(\eta))^\alpha (1 + \zeta)^\alpha \, \mathrm{OPT}(w^{\mathrm{pred}}) \, .$$

Hence, by choosing $\eta' = \zeta$,

$$\mathrm{OPT}'(\hat{w}^{\mathrm{online}}) \leqslant (1 + O(\eta))^\alpha (1 + \zeta)^{2\alpha} \, \mathrm{OPT}(w^{\mathrm{pred}}) + O(1/\zeta)^{4\alpha} \, \mathsf{err}_\eta(w^{\mathrm{real}}, w^{\mathrm{pred}}) \, .$$

It remains to upper bound $\mathrm{OPT}(w^{\mathrm{pred}})$ by $\mathrm{OPT}(w^{\mathrm{real}})$. Let $w = \mathrm{argmin}_{w \in N_\eta(w^{\mathrm{pred}})} \mathsf{err}(w, w^{\mathrm{real}})$ and so $\mathsf{err}_\eta(w^{\mathrm{real}}, w^{\mathrm{pred}}) = \mathsf{err}(w^{\mathrm{real}}, w)$. By again applying the arguments of Theorem 2, we have for any $\eta' > 0$

$$\mathrm{OPT}(w) \leqslant (1 + \eta')^\alpha \, \mathrm{OPT}(w^{\mathrm{real}}) + O(1/\eta')^\alpha \, \mathsf{err}(w^{\mathrm{real}}, w) \, .$$

Now consider an optimal schedule of $w$ subject to that for every time $t$ the job $w_t$ is scheduled within the interval $[t + \eta D, t + (1 - \eta)D]$. By Lemma 19, we have that this schedule has cost at most $(1 + O(\eta))^\alpha \, \mathrm{OPT}(w)$. Observe that this schedule for $w$ also defines a feasible schedule for $w^{\mathrm{pred}}$ since the time of any job is shifted by at most $\eta D$ in $w$. Hence, by again selecting $\eta' = \zeta$,

$$\mathrm{OPT}(w^{\mathrm{pred}}) \leqslant (1 + O(\eta))^\alpha \, \mathrm{OPT}(w)$$
$$\leqslant (1 + O(\eta))^\alpha \left( (1 + \zeta)^\alpha \, \mathrm{OPT}(w^{\mathrm{real}}) + O(1/\zeta)^\alpha \, \mathsf{err}_\eta(w^{\mathrm{real}}, w^{\mathrm{pred}}) \right)$$

Finally, by combining all inequalities, we get

$$\mathrm{OPT}'(\hat{w}^{\mathrm{online}}) \leqslant (1 + O(\eta))^{2\alpha} \left( (1 + \zeta)^{3\alpha} \, \mathrm{OPT}(w^{\mathrm{real}}) + O(1/\zeta)^{4\alpha} \, \mathsf{err}_\eta(w^{\mathrm{real}}, w^{\mathrm{pred}}) \right)$$

$\square$

### F.2.1 Proof of Lemma 22

The lemma is trivially true if there were no jobs that had remaining workloads to be assigned uniformly, i.e., if we always have $W = 0$ at Step 16 of NOISE-ROBUST($\mathcal{A}$). So suppose that there was at least one such job and consider the directed bipartite graph $G$ with bipartitions $A$ and $B$ defined as follows:

- $A$ contains a vertex for each component of $w^{\mathrm{real}}$ and $B$ contains one for each component of $w^{\mathrm{online}}$. In other words, $A$ and $B$ contain one vertex for each time unit.
- There is an arc from $i \in A$ to $j \in B$ if $|i - j| \leqslant \eta D$, that is, if $w_i^{\mathrm{real}}$ could potentially be assigned to $w_j^{\mathrm{online}}$.

- There is an arc from $j \in B$ to $i \in A$ if part of the workload of $w_i^{\text{real}}$ was assigned to $w_j^{\text{online}}$ by NOISE-ROBUST$(\mathcal{A})$, i.e., if $x_{ij} > 0$.

Now let $t$ be the *last* time step such that the online algorithm had to assign the remaining workload of $w_t^{\text{real}}$ uniformly. So, by selection, $t + \eta D$ is the last time step so that $w_{t+\eta D}^{\text{online}} > (1 + \zeta)w_{t+\eta D}^{\text{pred}}$. For $k \geqslant 0$, define the sets

$$A_k = \{i \in A : \text{the shortest path from } t \text{ to } i \text{ has length } 2k \text{ in } G\},$$
$$B_k = \{j \in B : \text{the shortest path from } t \text{ to } j \text{ has length } 2k + 1 \text{ in } G\}.$$

Here $t$ stands for the corresponding vertex in $A$. The set $A_k$ consists of those time steps, for which the corresponding jobs in $w^{\text{real}}$ have been moved in $w^{\text{online}}$ to the time slots in $B_{k-1}$ but not to any time slot in $B_{k-2}, B_{k-3}, \ldots, B_0$; and $B_k$ are all the time slots where the jobs corresponding to $A_k$ could have been assigned (but no job in $A_{k-1}, A_{k-2}, \ldots, A_0$ could have been assigned). By the selection of $t$, and the construction of $w^{\text{online}}$, these sets satisfy the following two properties:

**Claim 24.** *The sets $(A_k, B_k)_{k \geqslant 0}$ satisfy*

- *For any time step $j \in \bigcup_k B_k$ we have $w_j^{\text{online}} \geqslant (1 + \zeta)w_j^{\text{pred}}$.*

- *For any two time steps $i_k \in A_k$ and $i_\ell \in A_\ell$ with $k > \ell$, we have $i_k - i_\ell \leqslant 2\eta D(k - \ell + 2)$.*

*Proof of claim.* In the proof of the claim we use the notation $\ell(A_k)$ and $\ell(B_k)$ to denote the left-most (earliest) time step in $A_k$ and $B_k$, respectively. The proof is by induction on $k \geqslant 0$ with the following induction hypothesis (IH):

1. For any time step $j \in B_k$ we have $w_j^{\text{online}} \geqslant (1 + \zeta)w_j^{\text{pred}}$.

2. $B_0 = \{t - \eta D, \ldots, t + \eta D\}$ and for any (non-empty) $B_k$ with $k > 1$ we have $B_k = \{\ell(B_k), \ldots, \ell(B_{k-1}) - 1\}$ and $\ell(B_k) - \ell(B_{k-1}) \leqslant 2\eta D$.

The first part of IH immediately implies the first part of the claim. The second part implies the second part of the claim as follows: Any time step in $A_\ell$ has a time step in $B_\ell$ that differs by at most $\eta D$. Similarly, for any time step in $A_k$ there is a time step in $B_{k-1}$ at distance at most $\eta D$. Now by the second part of the induction hypothesis, the distance between these time steps in $B_{k-1}$ and $B_\ell$ is at most $(k - \ell + 1)2\eta D$.

We complete the proof by verifying the inductive hypothesis. For the base case when $k = 0$, we have $B_0 = \{t - \eta D, \ldots, t + \eta D\}$ by definition since $A_0 = \{t\}$. We also have that the first part of IH holds by the definition of NOISE-ROBUST$(\mathcal{A})$ and the fact that the overflow of job $w^{\text{real}}(t)$ was uniformly assigned to these time steps.

For the inductive step, consider a time step $i \in A_k$. By definition $w_i^{\text{real}}$ was assigned to a time step in $B_{k-1}$ but to no time step in $B_{k-2} \cup \ldots \cup B_0$. Now suppose toward contradiction that there is a time step $j \in A_{k-1}$ such that $j < i$. But then by the greedy strategy of NOISE-ROBUST$(\mathcal{A})$ (jobs are assigned left-to-right), we reach the contradiction that $w_i^{\text{real}}$ must have been assigned to a time step in $B_{k-2} \cup \ldots \cup B_0$ if $k \geqslant 2$ since then $w_j^{\text{real}}$ is assigned to a time step in $B_{k-2}$. For $k = 1$, we have $j = t$ and so all time steps in $B_0$ were full (with respect to capacity $(1 + \zeta)w^{\text{pred}}$) after $t$ was processed. Hence, in this case, $w_i^{\text{real}}$ could only be assigned to a time step in $B_0$ if it it had overflow that was uniformly assigned by NOISE-ROBUST$(\mathcal{A})$, which contradicts the selection of $t$.

We thus have that each time step in $A_k$ is smaller than the earliest time step in $A_{k-1}$. It follows that $B_k = \{\ell(B_k), \ldots, \ell(B_{k-1}) - 1\}$ where $\ell(B_k) = \ell(A_k) - \eta D$. The bound $\ell(B_k) - \ell(B_{k-1}) \leqslant 2\eta D$ then follows since, by definition, $\{\ell(A_k) - \eta D, \ldots, \ell(A_k) + \eta D\}$ must intersect $B_{k-1}$. This completes the inductive step for the second part of IH. For the first part, note that the job $w_{\ell(A_k)}^{\text{real}}$ was also assigned to $B_{k-1}$ by NOISE-ROBUST$(\mathcal{A})$. By the greedy left-to-right strategy, this only happens if the capacity of all time steps $B_k$ is saturated. $\square$

Now let $p$ be the smallest index such that $w^{\text{real}}(A_{p+1}) + w^{\text{real}}(A_{p+2}) \leqslant \zeta' \sum_{i=0}^{p} w^{\text{real}}(A_i)$ where we select $\zeta' = \zeta/10$. We have

$$\sum_{i=0}^{p+1} w^{\text{real}}(A_i) \geqslant \sum_{i=0}^{p} w^{\text{online}}(B_i) \geqslant (1+\zeta) \sum_{i=0}^{p} w^{\text{pred}}(B_i) \tag{6}$$

where the first inequality holds by the definition of the sets and the second is by the first part of the above claim. In addition, by the selection of $p$,

$$\sum_{i=0}^{p} w^{\text{real}}(A_i) \geqslant (1-\zeta') \sum_{i=0}^{p+2} w^{\text{real}}(A_i). \tag{7}$$

Now let $q = \max\{p - 4/(\zeta')^2, 0\}$. We claim the following inequality

$$\sum_{i=q}^{p} w^{\text{real}}(A_i) \geqslant (1-\zeta') \sum_{i=0}^{p} w^{\text{real}}(A_i). \tag{8}$$

The inequality is trivially true if $q = 0$. Otherwise, we have by the selection of $p$,

$$\begin{aligned}
\sum_{i=q}^{p} w^{\text{real}}(A_i) &= (1-\zeta') \sum_{i=q}^{p} w^{\text{real}}(A_i) + \zeta' \sum_{i=q}^{p} w^{\text{real}}(A_i) \\
&\geqslant (1-\zeta') \sum_{i=q}^{p} w^{\text{real}}(A_i) + \frac{(p-q)}{2} (\zeta')^2 \sum_{i=0}^{q-1} w^{\text{real}}(A_i) \\
&\geqslant (1-\zeta') \sum_{i=q}^{p} w^{\text{real}}(A_i) + 2 \sum_{i=0}^{q-1} w^{\text{real}}(A_i)
\end{aligned}$$

and so (8) holds.

We are now ready to complete the proof of the lemma. Let $w^*$ be a minimizer of the right-hand-side, i.e.,

$$w^* = \operatorname*{argmin}_{w \in N_\eta(w^{\text{real}})} \sum_i \left[ \left( w_i - w_i^{\text{pred}} \right)^+ \right]^\alpha$$

Divide the time steps of the instance into $T_1, B_{p+1}, T_2$ and $T_3$ where $T_1$ contains all time steps earlier than $\ell(B_{p+1})$, $T_2$ contains the time steps in $\cup_{i=0}^{p} B_i$, and $T_3$ contains the remaining time steps, i.e., those after $t + \eta D$. By the selection of $t$, we have $w_i^{\text{online}} \leqslant (1+\zeta) w_i^{\text{pred}}$ for all $i \in T_3$. We thus have that $\sum_i \left[ \left( w_i^{\text{online}} - (1+\zeta) w_i^{\text{pred}} \right)^+ \right]^\alpha$ equals

$$\sum_{i \in T_1} \left[ \left( w_i^{\text{online}} - (1+\zeta) w_i^{\text{pred}} \right)^+ \right]^\alpha + \sum_{i \in B_{p+1} \cup T_2} \left[ \left( w_i^{\text{online}} - (1+\zeta) w_i^{\text{pred}} \right)^+ \right]^\alpha.$$

We start by analyzing the second sum. The only jobs in $w^{\text{real}}$ that contribute to the workload of $w^{\text{online}}$ at the time steps in $B_{p+1} \cup T_2$ are by definition those corresponding to time steps in $A_0 \cup \ldots \cup A_{p+2}$. In the worst case, we have that $w^{\text{pred}}$ is 0 during these time steps and that the jobs in $w^{\text{real}}$ are uniformly assigned to the same $2\eta D + 1$ time steps. This gives us the upper bound:

$$\begin{aligned}
\sum_{i \in B_{p+1} \cup T_2} \left[ \left( w_i^{\text{online}} - (1+\zeta) w_i^{\text{pred}} \right)^+ \right]^\alpha &\leqslant \left( \frac{\sum_{i=0}^{p+2} w^{\text{real}}(A_i)}{2\eta D + 1} \right)^\alpha \cdot (2\eta D + 1) \\
&\leqslant (1+\zeta')^\alpha \left( \frac{\sum_{i=0}^{p} w^{\text{real}}(A_i)}{2\eta D} \right)^\alpha 2\eta D.
\end{aligned}$$

At the same time, combining (6) (7), and (8) give us

$$\sum_{i=q}^{p} w^{\text{real}}(A_i) \geqslant (1-\zeta')^2 (1+\zeta) \sum_{i=0}^{p} w^{\text{pred}}(B_i) \geqslant (1+\zeta/2) \sum_{i=0}^{p} w^{\text{pred}}(B_i).$$

By definition, the jobs in $w^{\text{real}}$ corresponding to time steps $\cup_{k=q}^{p} A_k$ can only be assigned to $w^{\text{online}}$ during time steps $T_2 = \cup_{k=0}^{p} B_k$. Therefore, as the difference between the largest time and smallest time in $\cup_{k=q}^{p} A_k$ is at most $2\eta D(p - q + 2)$ (second statement of the above claim) and thus the workload of those time steps can be assigned to at most $2\eta D(p - q + 4)$ time steps, we have

$$\sum_{i \in T_2} \left[ \left( w_i^* - w_i^{\text{pred}} \right)^+ \right]^\alpha \geqslant \left( \frac{\sum_{i=q}^{p} w^{\text{real}}(A_i) - \sum_{i=0}^{p} w^{\text{pred}}(B_i)}{(p - q + 4) \cdot 2\eta D} \right)^\alpha \cdot (p - q + 4) \cdot 2\eta D$$

$$\geqslant \left( c \cdot \zeta^3 \right)^\alpha \left( \frac{\sum_{i=0}^{p} w^{\text{real}}(A_i)}{2\eta D} \right)^\alpha \cdot 2\eta D$$

for an absolute constant $c$. It follows that

$$\sum_{i \in B_{p+1} \cup T_2} \left[ \left( w_i^{\text{online}} - (1 + \zeta) w_i^{\text{pred}} \right)^+ \right]^\alpha \leqslant \left( \frac{1 + \zeta'}{c\zeta^3} \right)^\alpha \sum_{i \in T_2} \left[ \left( w_i^* - w_i^{\text{pred}} \right)^+ \right]^\alpha .$$

We have thus upper bounded the sum on the left over time steps in $B_{p+1} \cup T_2$ by the sum on the right over only time steps in $T_2$. Since NOISE-ROBUST($\mathcal{A}$) does not assign the workload $w_i^{\text{real}}$ for $i \in T_1$ to $w^{\text{online}}$ on any of the time steps in $T_2$, we can repeatedly apply the arguments on the time steps in $T_1$ to show

$$\sum_{i \in T_1} \left[ \left( w_i^{\text{online}} - (1 + \zeta) w_i^{\text{pred}} \right)^+ \right]^\alpha \leqslant \left( \frac{1 + \zeta'}{c\zeta^3} \right)^\alpha \sum_{i \in T_1 \cup B_{p+1}} \left[ \left( w_i^* - w_i^{\text{pred}} \right)^+ \right]^\alpha ,$$

yielding the statement of the lemma.

## G  ROBUSTIFY for uniform deadlines

Here we provide the proofs of Claim 4, Claim 5, Claim 6.

**Claim 4.** *If $s$ is a feasible schedule for $(w^{\text{real}}, (1 - \delta)D, T)$ then $s^{(\delta)}$ is a feasible schedule for $(w^{\text{real}}, D, T)$.*

*Proof.* Since $s$ is a feasible schedule for $(w, (1 - \delta D), T)$, we have that

$$\int_{r_i}^{r_i+D} s_i^{(\delta)}(t)dt = \int_{r_i}^{r_i+D} \frac{1}{\delta D} \left( \int_{t-\delta D}^{t} s_i(t')dt' \right) dt = \int_{r_i}^{r_i+(1-\delta)D} s_i(t') \left( \int_{t'}^{t'+\delta D} \frac{1}{\delta D} dt \right) dt' = w_i.$$

$\square$

**Claim 5.** *The cost of schedule $s^{(\delta)}$ is not higher than that of $s$, that is,*

$$\int_0^T (s^{(\delta)}(t))^\alpha dt \leqslant \int_0^T (s(t))^\alpha dt.$$

*Proof.* The proof only uses Jensen's inequality in the second line and the statement can be calculated as follows.

$$\int_0^T \left( s^{(\delta)}(t) \right)^\alpha dt = \int_0^T \left( \frac{1}{\delta D} \int_{t-\delta D}^{t} s(t')dt' \right)^\alpha dt$$

$$\leqslant \int_0^T \frac{1}{\delta D} \left( \int_{t-\delta D}^{t} (s(t'))^\alpha dt' \right) dt$$

$$= \int_0^T (s(t'))^\alpha \left( \int_{t'}^{t'+\delta D} \frac{1}{\delta D} dt \right) dt'$$

$$= \int_0^T (s(t))^\alpha dt$$

$\square$

**Claim 6.** *Let $s$ be a feasible schedule for $(w^{\text{real}}, (1-\delta)D, T)$. Then $s_i^{(\delta)}(t) \leqslant \frac{1}{\delta} s_i^{\text{AVR}}(t)$.*

*Proof.* We have that

$$s_i^{(\delta)}(t) = \frac{1}{\delta D} \int_{t-\delta D}^{t} s_i(t')dt' \leqslant \frac{1}{\delta D} \int_{r_i}^{r_i+D} s_i(t')dt' = \frac{w_i}{\delta D} = \frac{s_i^{\text{AVR}}(t)}{\delta}.$$

$\square$

## H ROBUSTIFY for general deadlines

In this section, we discuss generalizations of our techniques to general deadlines. Recall that an instance with general deadlines is defined by a set $\mathcal{J}$ of jobs $J_j = (r_j, d_j, w_j)$, where $r_j$ is the time the job becomes available, $d_j$ is the deadline by which it must be completed, and $w_j$ is the work to be completed. For $\delta > 0$, we use the notation $\mathcal{J}^\delta$ to denote the instance obtained from $\mathcal{J}$ by shrinking the duration of each job by a factor $(1-\delta)$. That is, for each job $(r_j, d_j, w_j) \in \mathcal{J}$, $\mathcal{J}^\delta$ contains the job $(r_j, r_j + (1-\delta)(d_j - r_j), w_j)$.

Our main result in this section generalizes ROBUSTIFY to general deadlines.

**Theorem 25.** *For any $\delta > 0$, given an online algorithm for general deadlines that produces a schedule for $\mathcal{J}^\delta$ of cost $C$, we can compute online a schedule for $\mathcal{J}$ of cost at most*

$$\min\left\{ \left(\frac{1}{1-\delta}\right)^{\alpha-1} C, (2\alpha/\delta^2)^\alpha/2 \cdot \text{OPT} \right\},$$

*where* OPT *denotes the cost of an optimal schedule of $\mathcal{J}$.*

Since it is easy to design a consistent algorithm by just blindly following the prediction, we have the following corollary.

**Corollary 26.** *There exists a learning augmented online algorithm for the General Speed Scaling problem, parameterized by $\varepsilon > 0$, with the following guarantees:*

- Consistency: *If the prediction is accurate, then the cost of the returned schedule is at most $(1+\varepsilon)\text{OPT}$.*

- Robustness: *Irrespective of the prediction, the cost of the returned schedule is at most $O(\alpha^3/\varepsilon^2)^\alpha \cdot \text{OPT}$.*

*Proof of Corollary.* Consider the algorithm that blindly follows the prediction to do an optimal schedule of $\mathcal{J}^\delta$ when in the consistent case. That is, given the prediction of $\mathcal{J}$, it schedules all jobs that agrees with the prediction according to the optimal schedule of the predicted $\mathcal{J}^\delta$; the workload of the remaining jobs $j$ that were wrongly predicted is scheduled uniformly during their duration from release time $r_j$ to deadline $d_j$. In the consistent case, when the prediction is accurate, the cost of the computed schedule equals thus the cost $\text{OPT}(J^\delta)$ of an optimal schedule of $J^\delta$. Furthermore, we have by Lemma 19

$$\text{OPT}(\mathcal{J}^\delta) \leqslant \left(\frac{1}{1-\delta}\right)^{\alpha-1} \text{OPT},$$

where OPT denotes the cost of an optimal schedule to $\mathcal{J}$. Applying Theorem 25 on this algorithm we thus obtain an algorithm that is also robust. Specifically, we obtain an algorithm with the following guarantees:

- If prediction is accurate, then the computed schedule has cost at most $\left(\frac{1}{1-\delta}\right)^{2(\alpha-1)} \cdot \text{OPT}$.

- The cost of the computed schedule is always at most $(2\alpha/\delta^2)^\alpha/2 \cdot \text{OPT}$.

The corollary thus follows by selecting $\delta = \Theta(\varepsilon/\alpha)$ so that $1/(1-\delta)^{2(\alpha-1)} = 1 + \varepsilon$.

$\square$

We remark that one can also define "smooth" algorithms for general deadlines as we did in the uniform case. However, the prediction model and the measure of error quickly get complex and notation heavy. Indeed, our main motivation for studying the Uniform Speed Scaling problem is that it is a clean but still relevant version that allows for a natural prediction model.

We proceed by proving the main theorem of this section, Theorem 25.

**The procedure GENERAL-ROBUSTIFY.**   We describe the procedure GENERAL-ROBUSTIFY that generalizes ROBUSTIFY to general deadlines. Its analysis then implies Theorem 25. Let $\mathcal{A}$ denote the online algorithm of Theorem 25 that produces a schedule of $\mathcal{J}^\delta$ of cost $C$. To simplify the description of GENERAL-ROBUSTIFY, we fix $\Delta > 0$ and assume that the schedule $s$ output by $\mathcal{A}$ only changes at times that are multiples of $\Delta$. This is without loss of generality as we can let $\Delta$ tend to $0$. To simplify our calculations, we further assume that $\delta(d_j - r_j)/\Delta$ evaluates to an integer for all jobs $(r_j, d_j, w_j) \in \mathcal{J}$.

The time line is thus partitioned into time intervals of length $\Delta$ so that in each time interval either no job is processed by $s$ or exactly one job is processed at constant speed by $s$. We denote by $s(t)$ the speed at which $s$ processes the job $j(t)$ during the $t$:th time interval, where we let $s(t) = 0$ and $j(t) = \perp$ if no job was processed by $s$ (during this time interval).

To describe the schedule computed by GENERAL-ROBUSTIFY, we further divide each time interval into a *base* part of length $(1-\delta)\Delta$ and an *auxiliary* part of length $\delta\Delta$. In the $t$:th time interval, GENERAL-ROBUSTIFY schedules job $j(t)$ at a certain speed $s^{\text{base}}(t)$ during the base part, and a subset $\mathcal{J}(t) \subseteq \mathcal{J}$ of the jobs is scheduled during the auxiliary part, each $i \in J(t)$ at a speed $s_i^{\text{aux}}(t)$. These quantities are computed by GENERAL-ROBUSTIFY online at the start of the $t$:th time interval as follows:

- Let $s^{\text{aux}}(t) = \sum_{i \in \mathcal{J}(t)} s_i^{\text{aux}}(t)$ be the current speed of the auxiliary part and let $D_{j(t)} = d_{j(t)} - r_{j(t)}$ be the duration of job $j(t)$.

- If $s(t)/(1-\delta) \leqslant s^{\text{aux}}(t)$, then set $s^{\text{base}}(t) = s(t)/(1-\delta)$.

- Otherwise, set $s^{\text{base}}(t)$ so that

$$(1-\delta)\Delta s^{\text{base}}(t) + \left(s^{\text{base}}(t) - s^{\text{aux}}(t)\right)\delta^2 D_{j(t)} = s(t)\Delta \tag{9}$$

  and add $j(t)$ to $J(t), J(t+1), \ldots, J(t + \delta D_{j(t)}/\Delta - 1)$ with all auxiliary speeds $s_{j(t)}^{\text{aux}}(t), s_{j(t)}^{\text{aux}}(t+1), \ldots, s_{j(t)}^{\text{aux}}(t + \delta D_{j(t)}/\Delta - 1)$ set to $s^{\text{base}}(t) - s^{\text{aux}}(t)$.

This completes the formal description of GENERAL-ROBUSTIFY. Before proceeding to its analysis, which implies Theorem 25, we explain the example depicted in Figure 4. Schedule $s$, illustrated on the left, schedules a blue, red, and green job during the first, second, and third time interval, respectively. We have that $\delta/\Delta$ times the duration of the blue job and the red job are 3 and 4, respectively. GENERAL-ROBUSTIFY now produces the schedule on the right where the auxiliary parts are indicated by the horizontal stripes. When the the blue job is scheduled it is partitioned among the base part of the first interval and evenly among the auxiliary parts of the first, second and third intervals so that the speed at the first interval is the same in the base part and auxiliary part. Similarly, when the red job is scheduled, GENERAL-ROBUSTIFY splits it among the base part of the second interval and evenly among the auxiliary part of the second, third, fourth and fifth intervals so that the speed during the base part equals the speed at the auxiliary part during the second interval. Finally, the green job is processed at a small speed and is thus only scheduled in the base part of the third interval (with a speed increased by a factor $1/(1-\delta)$).

**Analysis.**   We show that GENERAL-ROBUSTIFY satisfies the guarantees stipulated by Theorem 25. We first argue that GENERAL-ROBUSTIFY produces a feasible schedule to $\mathcal{J}$. During the $t$:th interval, the schedule $s$ computed by $\mathcal{A}$ processes $\Delta \cdot s(t)$ work of job $j(t)$. We argue that GENERAL-ROBUSTIFY processes the same amount of work from this time interval. At the time when this interval is considered by GENERAL-ROBUSTIFY, there are two cases:

- If $s(t)/(1-\delta) \leqslant s^{\text{aux}}(t)$ then $s^{\text{base}}(t) = s(t)/(1-\delta)$ so GENERAL-ROBUSTIFY processes $(1-\delta)\Delta s(t)/(1-\delta) = s(t)\Delta$ work of $j(t)$ during the base part of the $t$:th time interval.

Figure 4: Given the schedule on the left, GENERAL-ROBUSTIFY produces the schedule on the right.

- Otherwise, we have that GENERAL-ROBUSTIFY processes $(1-\delta)\Delta s^{\text{base}}(t)$ of $j(t)$ during the base part of the $t$:th time interval and $\delta\Delta\left(s^{\text{base}}(t) - s^{\text{aux}}(t)\right)$ during the auxiliary part of each of the $\delta D_{j(t)}/\Delta$ time intervals $t, t+1, \ldots, t+\delta D_{j(t)}/\Delta - 1$. By the selection (9), it thus follows that GENERAL-ROBUSTIFY processes all work $s(t)\Delta$ from this time interval. in this case as well.

The schedule of GENERAL-ROBUSTIFY thus completely processes every job. Furthermore, since each job is delayed at most $\delta D_{j(t)}$ time steps we have that it is a feasible schedule to $\mathcal{J}$ since we started with a schedule for $\mathcal{J}^{\delta}$, which completes each job $j$ by time $r_j + (1-\delta)D_j$. It remains to prove the robustness and soundness guarantees of Theorem 25

**Lemma 27** (Robustness). GENERAL-ROBUSTIFY *computes a schedule of cost at most* $(2\alpha/\delta^2)^{\alpha}/2 \cdot$ OPT.

*Proof.* By the definition of the algorithm we have, for each time interval, that the speed of the base part is at most the speed of the auxiliary part. Letting $s^{\text{base}}(t)$ and $s^{\text{aux}}(t)$ denote the speed of the base and auxiliary part of the $t$:th time interval, we thus have

$$\sum_t \left((1-\delta)s^{\text{base}}(t)^{\alpha} + \delta s^{\text{aux}}(t)^{\alpha}\right) \leqslant \sum_t s^{\text{aux}}(t)^{\alpha}.$$

Now we have that the part of a job $j$ that is processed during the auxiliary part of a time interval has been uniformly assigned to at least $\delta^2 D_j$ time steps. It follows that the speed at any auxiliary time interval is at most $1/\delta^2$ times the speed at that time of the AVERAGE RATE heuristic (AVR). The lemma now follows since that heuristic is known [17] to have competitive ratio at most $(2\alpha)^{\alpha}/2$. $\square$

**Lemma 28** (Consistency). GENERAL-ROBUSTIFY *computes a schedule of cost at most* $\left(\frac{1}{1-\delta}\right)^{\alpha-1} \cdot C$ *where $C$ denotes the cost of the schedule $s$ computed by $\mathcal{A}$.*

*Proof.* For $t \geqslant 0$, let $h^{(t)}$ be the schedule that processes the workload during the first $t$ time intervals as in the schedule computed by GENERAL-ROBUSTIFY, and the workload of the remaining time intervals is processed during the base part of that time interval by increasing the speed by a factor $1/(1-\delta)$. Hence, $h^{(0)}$ is the schedule that processes the workload of all time intervals during the base part at a speed up of $1/(1-\delta)$, and $h^{(\infty)}$ equals the schedule produced by GENERAL-ROBUSTIFY. By definition, the cost of $h^{(0)}$ equals $\left(\frac{1}{1-\delta}\right)^{\alpha}(1-\delta) \cdot C$ and so the lemma follows by observing that for every $t \geqslant 1$ the cost of $h^{(t)}$ is at most the cost of $h^{(t-1)}$. To see this consider the two cases of GENERAL-ROBUSTIFY when considering the $t$:th time interval:

- If $s(t)/(1-\delta) \leqslant s^{\text{aux}}(t)$ then GENERAL-ROBUSTIFY processes all the workload during the base part at a speed of $s^{\text{base}}(t) = s(t)/(1-\delta)$. Hence, in this case, the schedules $h^{(t)}$ and $h^{(t-1)}$ processes the workload of the $t$:th time interval identically and so they have equal costs.

- Otherwise, GENERAL-ROBUSTIFY partitions the workload of the $t$:th time interval among the base part of the $t$:th interval and $\delta D_{j(t)}/\Delta$ many auxiliary parts so that the speed at each of these parts is strictly less than $s(t)/(1-\delta)$. Hence, since $h^{(t)}$ processes the workload of the $t$:th time interval at a lower speed than $h^{(t-1)}$ we have that its cost is strictly lower if $\alpha > 1$ (and the cost is equal if $\alpha = 1$).

$\square$

## I  Additional Experiments

In this section we further explore the performance of LAS algorithm for different values of the parameter $\alpha$. We conduct experiments on the login requests of *BrightKite* using the same experimental setup used in Section 4. The results are summarized in Table 2. In every column the average competitive ratios of each algorithm for a fixed $\alpha$ are presented. We note that, as expected, higher values of $\alpha$ penalize heavily wrong decisions deteriorating the competitive ratios of all algorithms. Nevertheless, LAS algorithm consistently outperforms AVR and OA for all different values of $\alpha$.

Table 2: Real dataset results with different $\alpha$ values

| Algorithm | $\alpha = 3$ | $\alpha = 6$ | $\alpha = 9$ | $\alpha = 12$ |
|---|---|---|---|---|
| AVR | 1.365 | 2.942 | 7.481 | 21.029 |
| OA | 1.245 | 2.211 | 4.513 | 9.938 |
| LAS, $\varepsilon = 0.8$ | 1.113 | 1.576 | 2.806 | 7.204 |
| LAS, $\varepsilon = 0.01$ | 1.116 | 1.598 | 2.918 | 8.055 |

The timeline was discretized in chunks of ten minutes and $D$ was set to 20.