[Reviews · NeurIPS 2020]

Review 1

Summary and Contributions: This submission considers the uniform speed scaling problem, which is an optimization problem to minimize the energy consumption by adjusting speed of processing unit under the constraint that all given jobs are processed before their deadlines. Algorithms for this optimization problems have been investigated in the previous studies both in the offline and the online setting, where the offline setting means that all the inputs of the problem are given to the algorithm and the online setting means that the workload of each job is not given to the algorithm in advance and it is revealed when the process of the job starts. For the online setting, algorithms with provable competitive ratios are known. The submission considers a new setting of the uniform speed scaling problem. It is almost same as the online setting, but the prediction on the workloads of jobs are also given. This prediction may not be correct, and the correct workloads are given as in the usual online setting. The submission presents a bound on the competitive ratio of an algorithm, which depends on the prediction error of the workload.

Strengths: - A new interesting setting of the online uniform speed scaling problem. The usual online setting (no information on the workload is available before the process of the job starts) is too restrictive, and the proposed setting is more realistic and useful. - The paper presents a theoretically provable performance guarantee of the proposed algorithm.

Weaknesses: - This is a purely algorithm paper. The result is on a kind of sensitive analysis of the algorithm. Although this result may be useful when it is combined with machine learning, it is more suitable to be presented in an algorithm conference. - The definition of the prediction error is artificial by a technical reason. I know no machine algorithm which gives a bound on this kind of prediction error.

Correctness: The analysis seems correct although I do not check all the details. The experimental setting is not clear. In particular, it does not explain how the predictions are used in the baseline online algorithms. Thus I am not sure the experiments are fair.

Clarity: The paper is written well.

Relation to Prior Work: Relation to the prior work is discussed appropriately.

Reproducibility: No

Additional Feedback: I read the authors' feedback and my opinion hasn't changed.


Review 2

Summary and Contributions: In the energy minimization via speed scaling problem, one is given a single machine (server) that can process jobs at variable speeds. The energy consumed by the machine at any time t is given by s(t)^alpha for some alpha>1 where s(t) refers to the speed of the machine at time t. The goal is to design an algorithm that processes all jobs within their deadlines and minimizes the total energy spend. This is a well studied problem and tight offline and online algorithms are known in this setting. The authors propose a learning-augmented online algorithm for this problem where the algorithm is provided with certain predictions regarding the workloads arriving at each time step. The goal is to design an algorithm that is (almost) optimal if the predictions are (almost) correct and still guarantee good worst case performance. The paper makes a number of interesting contributions. First, even in the standard setup without predictions, they show that the classical AVR algorithm (Yao) yields a 2^alpha approximation in the special case of uniform deadlines. The main result follows in two steps - first, the authors propose a new algorithm LAS-Trust that utilizes predictions and is consistent but not robust, i.e. it performs well when predictions are good. Second, the authors show a general technique to make any algorithm robust (even with general deadlines).

Strengths: The paper provides a novel learning augmented algorithm for a fundamental scheduling problem. The paper makes non-trivial technical contributions and is likely to spark further research in this area. At first glance, the error model introduced by the authors seems very brittle, but the authors take care to strengthen it in the appendix by allowing the predictions to be ‘shifted’.

Weaknesses: No obvious weaknesses. The paper considers an interesting problem and makes non-trivial contributions.

Correctness: Yes

Clarity: Yes

Relation to Prior Work: Yes

Reproducibility: Yes

Additional Feedback: Other comments: Line 44. typo; “well-funded” -> “well-founded”?


Review 3

Summary and Contributions: This paper studies an online scheduling problem with power constraint that's a polynomial of the workload. It gives an online algorithm with provable guarantees (as well as a lower bound in this model), and also analyzes the consistentness and robustness of the proposed routine.

Strengths: Online scheduling is quite intricate, and significant work was needed to get to an algorithm with provable guarantees.The paper also experimentally evaluates their algorithm on both synthetic and real data sets.

Weaknesses: The paper reads more like an online algorithms paper rather than a learning paper. While there was a brief discussion of the ability to incorporate arbitrary `downstream' schedulers, this connection was not expanded upon, and not very clear to me. On the experimental side, the workload (logins to a website) seems to be more close to a database application. The role of the \alpha parameter in performance is also not explicitly justified: if possible I'd liked to have seen it being verified experimentally.

Correctness: I believe the correctness of the guarantees, and find the experimental method for evaluating the scheduling algorithm reasonable. On the other hand, the dependence between workload and power consumption might differ for different applications: I'd prefer to see some experimental justifications of this model.

Clarity: While the motivation and context of the problem were clearly stated, it took some effort to figure out the formal definition of the problem as well as result. There are also a number of typos, e.g. on line 44, `funded' -> 'founded'.

Relation to Prior Work: Yes, the paper clearly discusses its improvements over previous algorithms.

Reproducibility: Yes

Additional Feedback: This result is a solid result in scheduling / online algorithms. While it may have connections with optimizing the performances of ML systems/algorithms, I'd like to see more explicit discussions of such connections in the paper. Per the feedback provided by the authors, I'm more convinced about the relevance of this result, and more generally, of the approach taken.


Review 4

Summary and Contributions: This paper studies the classical scheduling question of speed scaling with a new perspective -- one of learning-augmented online algorithms. In particular, instead of a purely worst-case view, the paper assumes the online algorithm has access to a noisy prediction of the instance and then shows how to use this model to develop an efficient black-box algorithm. Interestingly, the paper also provides a lower bound that shows that it is impossible to both be optimal when predictions are correct and "robust" to error at the same time.

Strengths: This paper brings the growing literature on learning-augmented online algorithms to a new setting -- speed scaling. Speed scaling is a classical problem where worst-case results tend to be particularly pessimistic and so it is ripe for improvement with this sort of beyond worst-case analysis. The proposed prediction model is interesting and novel. It is likely to be used in follow-up work in the context of other online algorithms.

Weaknesses: The design of the algorithms, while interesting theoretically, seems not particularly practical. In particular, the fact that LAS uses predictions of the full instance does not make sense in practice since predictions of the far future are likely to be very noisy and, in most situations, a much more limited prediction window is used in practice. The paper focuses on the simplest form of speed scaling, leaving more complex extensions to the appendix and not evaluating them numerically. The paper would be strengthened if the focus was on more general settings of the problem.

Correctness: I reviewed the provided proofs and did not find any issues.

Clarity: The paper is well written and clearly organized.

Relation to Prior Work: The paper provides a brief description of related work on speed scaling but misses some work that may be interesting to include. In particular, there is a literature on speed scaling in stochastic settings that is relevant, since the models can be viewed as providing stochastic predictions about the workload. In that context, the issue of robustness has also been considered, e.g., Optimality, fairness, and robustness in speed scaling designs by Andrew et al. and the references therein. Additionally, there is considerable work on predictions in related online problems that may provide additional context for the prediction model here. An example is the context of online optimization, where a variety of prediction models have been considered, e.g., see Online Optimization in Cloud Resource Provisioning: Predictions, Regrets, and Algorithms by Comden et al and the references therein

Reproducibility: Yes

Additional Feedback: Thank you for the author response. I have given the section in the appendix a careful read and it is interesting how you do the extension to allow the creation of independent segments. If the paper is accepted, I encourage you to include some discussion of these extensions in the body along with the discussion of additional related work.

[Author Response · NeurIPS 2020]

We thank the reviewers for their valuable feedback. In two of the reviews the paper's topic and its connection to NeurIPS was discussed. While the paper is clearly algorithmic, we believe that it is a good fit for NeurIPS. It is in the new and rapidly growing field of learning-augmented online algorithms that use predictions about the future, an important application of machine learning. We see it as interdisciplinary research with great potential, in particular, because some of the techniques might not be typical in this area. Numerous papers of this field have been presented at NeurIPS and other conferences of similar scope, see for example [6,7,12,13,14] in our literature review.

**Reviewer #1.** The baseline online algorithms do not use the predictions at all. Our algorithm is the first to consider this prediction setting. Hence, it is impossible to say that our algorithm is better or worse than those algorithms. The purpose of the experiment is not to make such a comparison, but to empirically verify the message of the theoretical analysis. Namely, we understand that indeed with a good prediction the algorithm is superior to online algorithms that do not use the prediction. Another crucial property of the algorithm is that it still performs reasonably well, if the prediction is very bad. If the prediction is misleading, it should intuitively be damaging to the algorithm to make any use of it. Since we consider both settings (good and bad predictions) we believe that our experimental setup is fair.

Regarding the measure of prediction error and that the reviewer knows no ML algorithm that gives such guarantees: Indeed, one of the strengths of the considered model is that the algorithm makes no assumptions on the guarantees of the prediction but is guaranteed to work well if predictions are good and be comparable to worst-case guarantees if the predictions are bad. We think that the considered measures of error is the most natural one except possibly for the power of $\alpha$ which we show is necessary.

**Reviewer #2.** No objections.

**Reviewer #3.** The parameter $\alpha$ is problem specific. If one is to consider energy consumption of processors in relation to its speed, the function is cubic, that is, $\alpha = 3$. This is arguably the most important case, which is why we have focused on it in the experiments. We refer to [Speed Scaling to Manage Energy and Temperature, Bansal, Kimbrel, and Pruhs] for a thorough justification of the model. We have run similar tests with other choices of $\alpha$ and observed the same behavior on the artificial data and a much better performance of LAS with respect to the classical online algorithms in the real data setting for larger $\alpha$. We will include those in the final version.

The ability to incorporate 'downstream' schedulers refers to our robustification methods (in section 3.2 and appendix H for general deadlines). Our methods can be used to make any schedule robust in a black-box manner (i.e. without making any assumption on the schedule) at a small multiplicative cost.

The data set we used was previously used in the evaluation of learning augmented algorithms (as we note in the experimental section) and we think that access pattern to a website is a good test bed for server workload.

**Reviewer #4.** Section D in the appendix deals with precisely the prediction setting proposed. Namely, we consider the predictions in the far future to be less reliable and as time advances the prediction is adjusted. The take-away of this section is that it is sensible to chop the timeline into small segments and to schedule each segment independently, which turns the problem into the one considered in the main body. We also show with strong lower bounds that one cannot do much better than that. Hence, the interesting core of the problem lies in the case we consider in the main body. Indeed, the setting in the main body is somewhat restrictive, but there is an argument to be made for having a clean and self-contained study in the main body. We feel that the more general algorithms could not be discussed appropriately under the given space constraints, hence we decided to move them to the appendix.

We thank the reviewer for providing additional related literature. We will include it in the final version of the paper.

[Meta-Review · NeurIPS 2020]

This paper tackles the uniform speed scaling problem in the emerging paradigm of algorithms with predictions. The authors give an interesting and intricate analysis and evaluate the performance of their algorithm experimentally.